# Exceptional non-Hermitian topological edge mode and its application to active matter

Kazuki Sone [1✉], Yuto Ashida[1,3] & Takahiro Sagawa[1,2]

Topological materials exhibit edge-localized scattering-free modes protected by their non-trivial bulk topology through the bulk-edge correspondence in Hermitian systems. While topological phenomena have recently been much investigated in non-Hermitian systems with dissipations and injections, the fundamental principle of their edge modes has not fully been established. Here, we reveal that, in non-Hermitian systems, robust gapless edge modes can ubiquitously appear owing to a mechanism that is distinct from bulk topology, thus indicating the breakdown of the bulk-edge correspondence. The robustness of these edge modes originates from yet another topological structure accompanying the branchpoint singularity around an exceptional point, at which eigenvectors coalesce and the Hamiltonian becomes nondiagonalizable. Their characteristic complex eigenenergy spectra are applicable to realize lasing wave packets that propagate along the edge of the sample. We numerically confirm the emergence and the robustness of the proposed edge modes in the prototypical lattice models. Furthermore, we show that these edge modes appear in a model of chiral active matter based on the hydrodynamic description, demonstrating that active matter can exhibit an inherently non-Hermitian topological feature. The proposed general mechanism would serve as an alternative designing principle to realize scattering-free edge current in non-Hermitian devices, going beyond the existing frameworks of non-Hermitian topological phases.

[1] Department of Applied Physics, The University of Tokyo, 7-3-1 Hongo, Bunkyo-ku, Tokyo 113-8656, Japan. [2] Quantum-Phase Electronics Center (QPEC), The University of Tokyo, 7-3-1 Hongo, Bunkyo-ku, Tokyo 113-8656, Japan. [3] Present address: Department of Physics, The University of Tokyo, 7-3-1 Hongo, Bunkyo-ku, Tokyo 113-0033, Japan. ✉email: sone@noneq.t.u-tokyo.ac.jp

Since topological materials exhibit robust scattering-free currents along the edges of samples, the notion of topology has attracted much interest both in fundamental physics and in device engineering. The first discovery of such a topological material dates back to the integer quantum Hall effect[1], where it has been established that the gapless edge modes precisely correspond to the bulk topological number[2]. This phenomenon has revealed a fundamental principle known as the bulk-edge correspondence. Nowadays, the topological materials have been found in much broader situations especially in the presence of symmetry, such as the time-reversal symmetry in topological insulators[3,4]. In such situations, the bulk-edge correspondence is still valid and predicts the presence or absence of robust edge modes[4], from which a periodic table has been obtained[5]. These discoveries have opened up a stream of material designs on the basis of the bulk band topology[4,6].

Although the conventional notion of topological materials is based on Hermitian Hamiltonians, effective Hamiltonians can become non-Hermitian in nonconservative systems including both quantum and classical ones, such as photonics[7–10], ultracold atoms[11,12], optomechanics[13,14], electronic circuits[15,16], mechanical lattices[17,18], and biophysical systems[19]. For example, in photonic systems, non-Hermiticity can be introduced by engineering optical gain and loss through semiconductor amplifiers or acoustic modulators. Classification of non-Hermitian topological materials has been explored in terms of bulk band topology[10,20–32], and a periodic table has been proposed in the same spirit as in the Hermitian case[23,30,31]. However, the bulk-edge correspondence is more subtle in non-Hermitian systems than in Hermitian systems, as a hitherto unknown non-Hermitian effect may protect unpredicted edge modes or destabilize edge modes in topologically nontrivial systems.

In this article, we reveal a ubiquitous mechanism for realizing robust gapless edge modes, which emerge independently of the bulk topology and instead are protected in an unconventional manner unique to non-Hermitian systems. This indicates that the bulk-based classification cannot conclusively predict the existence or absence of edge modes in the non-Hermitian case, thus implying the breakdown of the bulk-edge correspondence. In the conventional topologically nontrivial systems, the proposed mechanism can further stabilize the gapless edge modes, even against symmetry-breaking disorder. These edge modes inherently exhibit large positive imaginary parts of the eigenenergies and thus are naturally applicable to topological insulator laser[33–35], where the amplified unidirectional wave packet propagates along the edge of the sample. We demonstrate the emergence of the proposed gapless edge modes and the lasing wave packets by analyzing the prototypical tight-binding models.

Our edge modes owe their robustness to the distinct topological structure of exceptional points (EPs), and thus here we term these modes as exceptional edge modes. The EP[36] is a singular point in the parameter space at which two or more eigenvectors and eigenvalues coalesce and a parameterized Hamiltonian becomes nondiagonalizable. The EP is unique to non-Hermitian systems and induces intriguing phenomena, such as interchanging eigenvectors after encircling an EP[37], coherent perfect absorption[38], and unidirectional invisibility[39]. The existence of EPs is supported by the nontrivial topology of the branchpoint singularity in intersecting Riemann surfaces around them.[37,40] In one-dimensional systems, such as the edge modes of two-dimensional bulk systems, the emergence of EPs can be guaranteed by satisfying certain symmetries, including the *PT* symmetry, the *CP* symmetry, the pseudo-Hermiticity, and the chiral symmetry[41–43]. EPs can disappear if either the symmetry is broken or a pair of EPs coalesce; the latter is reminiscent of the pair-annihilation of Weyl points[44,45] in Hermitian systems. We

discover a general mechanism that EPs join two edge dispersions like glue and make them robust against disorder, which cannot be predicted by the existing periodic tables of topological phases[23,30,31].

Furthermore, we explicitly show the existence of exceptional edge modes in a more realistic system based on active matter[46], which is a collection of self-propelled particles and has recently attracted much interest as a useful platform to study biological and out-of-equilibrium physics. Recent studies[47–52] have explored the existence of the edge modes in active matter protected by the bulk topology. Some of them[49,50,52] have utilized chiral active matter, which moves in a circular path or self-rotates. Chiral active matter has been experimentally realized, for example, in bacteria[53] and artificial L-shaped particles[54]. The hydrodynamics[55,56] and the phase separation[57] of chiral active matter have also been analyzed in recent studies. The effective Hamiltonian of the linearized hydrodynamic equations in active matter is, in general, non-Hermitian because of inherent dissipations and energy injections therein. We demonstrate that this type of non-Hermitian chiral active matter provides an ideal platform to experimentally realize the proposed exceptional edge modes.

## Results

**Exceptional edge modes in two-layered non-Hermitian Bernevig-Hughes-Zhang model**. We first construct and analyze a minimal tight-binding model. For a Hermitian Hamiltonian $H$, time-reversal symmetry means that there exists a unitary operator $T$ satisfying $TH(\mathbf{k})T^{-1} = H^*(-\mathbf{k})$, where $H(\mathbf{k})$ is the Bloch Hamiltonian constructed from $H$[4]. The definition of time-reversal symmetry can be extended to non-Hermitian systems and it has been pointed out[21,23,30,31] that there are two types of time-reversal symmetry, i.e., $TH(\mathbf{k})T^{-1} = H^*(-\mathbf{k})$ and $TH(\mathbf{k})T^{-1} = H^T(-\mathbf{k})$, which are equivalent in Hermitian systems while not in non-Hermitian cases. One can construct the conventional time-reversal-symmetric topological insulator by coupling a Chern insulator with its time-reversal counterpart[3,58]. The bulk bands of a time-reversal-symmetric insulator are topologically characterized by the $\mathbb{Z}_2$ index[3,4], which corresponds to the parity of the number of the edge modes across the Fermi energy. If we construct a time-reversal-symmetric system from Chern insulators with even numbers of edge modes, we obtain a topologically trivial bulk. However, we reveal that such a trivial bulk can still accompany robust gapless edge modes by introducing non-Hermitian coupling between the two Chern insulators (see Fig. 1a).

To construct the minimal model for demonstrating the emergence of such edge modes, we consider the two-layered Qi-Wu-Zhang (QWZ) model, $H_0 = I_2 \otimes H_{\text{QWZ}} + c\sigma_x \otimes I_2$, which exhibits two chiral modes per edge in the bulk energy gap. Here, $H_{\text{QWZ}}$ is the Hamiltonian of the QWZ model[59], which can be described as $H_{\text{QWZ}}(\mathbf{k}) = \sin k_x \sigma_x + \sin k_y \sigma_y + (u + \cos k_x + \cos k_y)\sigma_z$ in the wavenumber space (see Supplementary Methods for the real-space description). Here, $I_2$ is the $2 \times 2$ identity matrix and $\sigma_i$ is the $i$th component of the Pauli matrices. Also, we assume that $c$ is real, and thus the Hamiltonian is still Hermitian. By coupling $H_0$ and its time-reversal counterpart $H_0^*$ with a non-Hermitian term, $i\Sigma = i(\beta + \beta')I_2 \otimes \sigma_x/2 + i(\beta - \beta')\sigma_z \otimes \sigma_x/2$ with $\beta$, $\beta'$ being real parameters, we obtain the following non-Hermitian Hamiltonian

$$H = \begin{pmatrix} H_0 & i\Sigma \\ i\Sigma & H_0^* \end{pmatrix}. \tag{1}$$

We note that this model resembles the Bernevig-Hughes-Zhang model[58] but differs from it since our model has two layers of the

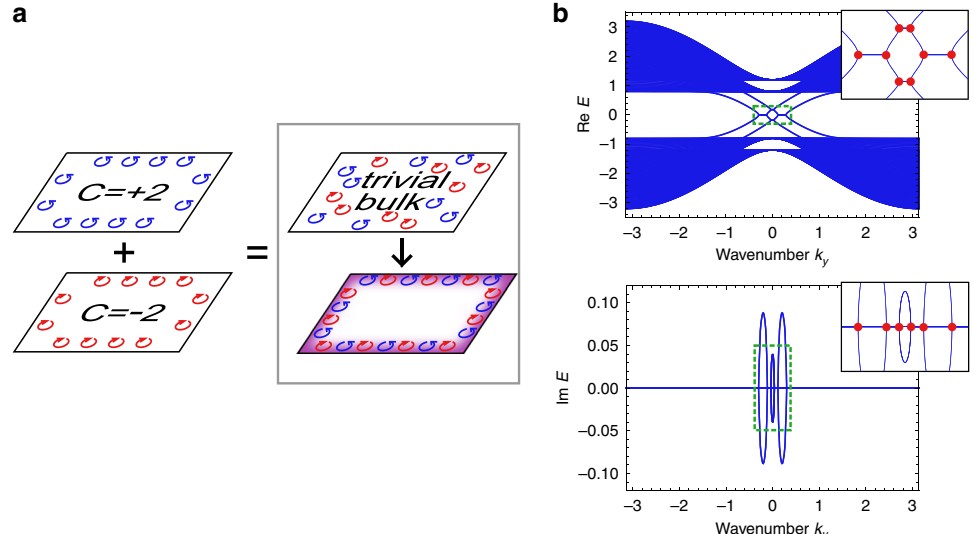

**Fig. 1 Schematic figure and prototypical band structure of exceptional edge modes. a** The system consists of two Chern insulators, which have the Chern numbers with the same absolute values and the opposite signs. If we consider Chern insulators with even Chern numbers, the bulk of the combined system becomes topologically trivial. However, when we use a non-Hermitian coupling to combine two Chern insulators, robust gapless modes can appear at the edge of the sample. **b** Two-layered non-Hermitian Bernevig-Hughes-Zhang model is considered to demonstrate the prototypical band structure of exceptional edge modes. We numerically calculate the edge band structure of the 1 × 50 ribbon-shaped system under the open boundary condition in the $x$ direction and the periodic boundary condition in the $y$ direction. The parameters used are $u = -1$, $c = 0.2$, $\beta = 0.14$, $\beta' = 0.06$, and $\gamma = 0.05$. Four gapless bands per edge exist in the bulk energy gap (they are doubly degenerate) and thus imply the topologically trivial bulk of the system. In the edge bands, we also find exceptional points (EPs) (indicated by red points in the insets), which are the wavenumbers at which the edge eigenstates coalesce and the Hamiltonian becomes nondiagonalizable. We can confirm that the imaginary parts of the eigenenergies appear from the EPs. These EPs play the role of the glue of the edge modes and thus prevent gap opening by perturbations or disorders, which is shown in Fig. 3.

QWZ model and two other layers of the time-reversal QWZ model, which are coupled by the non-Hermitian term. Furthermore, as this Hamiltonian has the pseudo-Hermiticity defined as $\eta H(\mathbf{k})\eta^{-1} = H^\dagger(\mathbf{k})$ which can lead to another topological classification characterized by the $\mathbb{Z}$ invariant[30,31], we add a Hermitian coupling and consider the Hamiltonian $H' = H + \gamma\sigma_x \otimes \sigma_y \otimes \sigma_x$, to break the pseudo-Hermiticity. The additional Hermitian coupling corresponds to the spin coupling in condensed matter and thus can open an energy gap in the conventional trivial insulator. We can confirm that this Hamiltonian has time-reversal symmetry $TH'(\mathbf{k})T^{-1} = H'^*(-\mathbf{k})$, and thus have to consider $\mathbb{Z}_2$ indices as in Hermitian systems (see Supplementary Note 10). Below we focus on the parameter regimes in which the bulk bands are trivial in the conventional sense, i.e., the number of the edge modes in $H_0$ is even.

To reveal the existence of robust edge modes, we calculate the band structure of our model with open (periodic) boundaries in the $x$ ($y$) direction. Figure 1b shows the band structure for the wavenumber in the $y$ direction. There, gapless edge bands exist in the bulk energy gap and they accompany EPs, where both the eigenenergies and the eigenstates coalesce. The EPs act as a glue that holds the edge band structures together and thus stabilize the existence of exceptional edge modes. This gluing is reminiscent of the branchpoint structure in non-Hermitian bulk bands[42,60], which remains until the EPs coalesce. In general models including the present one, the edge modes between two EPs exhibit the large imaginary parts of the eigenenergies, whereas all the bulk modes can have zero imaginary parts of the eigenenergies. As discussed below, this property finds a possible application to realize a topological insulator laser[33,34].

To explicitly demonstrate that the appearance of EPs is independent of the bulk topology and thus violates the bulk-edge correspondence, we numerically calculate the edge band structures for different strengths of the non-Hermitian coupling. By modifying the strengths of the non-Hermitian coupling $\beta$, $\beta'$, we can control the existence of edge modes and EPs in the bulk gap as shown in Fig. 2. On the other hand, during this modification, the bulk energy gap remains open. Therefore, the bulk topology should remain trivial at arbitrary strength of the non-Hermitian coupling and thus have no relation to the exceptional edge modes. This result indicates that while the bulk band topology still can predict the existence of the ordinary edge modes without EPs, it fails to predict the existence of robust exceptional edge modes.

Although we have concentrated on time-reversal-symmetric systems so far, time-reversal symmetry is not the prerequisite for realizing exceptional edge modes. If the sum of the Chern numbers of the bulk bands below the energy gap is zero, a system without relevant symmetries cannot exhibit gapless edge modes protected by bulk topology. However, combining the topological systems with the opposite Chern numbers by the non-Hermitian coupling, we can obtain not only a trivial bulk but also robust exceptional edge modes.

We can also realize exceptional edge modes in topologically nontrivial systems. We construct the non-Hermitian Bernevig-Hughes-Zhang model,

$$H = \begin{pmatrix} H_{\mathrm{QWZ}} & i\Sigma' \\ i\Sigma' & H_{\mathrm{QWZ}}^* \end{pmatrix}, \tag{2}$$

where $i\Sigma'$ is the non-Hermitian coupling $i\Sigma' = i\beta\sigma_x$, and $\beta$ is real. This model satisfies the time-reversal symmetry and associates with a nontrivial $\mathbb{Z}_2$ invariant. We calculate the edge band structure and confirm the existence of the exceptional edge modes (see Supplementary Note 2). We note that the exceptional edge modes can also exist robustly against time-reversal-symmetry-breaking disorder and thus can be more advantageous than conventional edge modes.

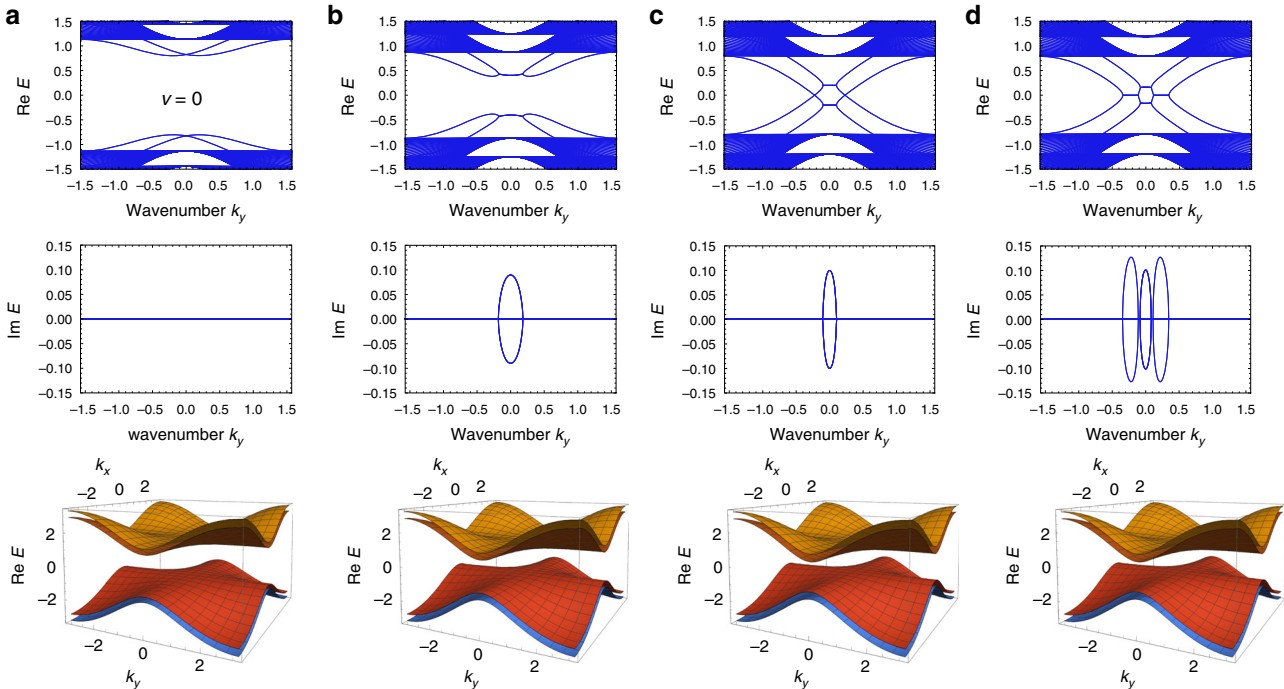

**Fig. 2 Emergent exceptional edge modes through a non-Hermitian coupling. a–d** Non-Hermiticity is increased from **a** to **d**. The upper two figures show the real and imaginary parts of the edge dispersions. The lowest figure represents the bulk band structure of the homogeneous system under the periodic boundary conditions. We set the parameters $u = -1$, $c = 0.2$, and $\gamma = 0.8$ throughout the calculations in this Figure. **a** Without non-Hermitian coupling ($\beta = \beta' = 0$), there is a large energy gap even in the edge band structure. Therefore, the bulk topology is trivial in this Hermitian system. **b** With increased non-Hermiticity ($\beta = 0.81$, $\beta' = 0.63$), the upper and lower edge dispersions approach to each other. **c** At the critical strength of the non-Hermitian coupling ($\beta = 0.9$, $\beta' = 0.7$), the upper and lower edge dispersions coalesce. However, there are no gap closings in the bulk band structure. Thus, the bulk topology should remain trivial. **d** With a stronger non-Hermitian coupling than the critical value ($\beta = 0.909$, $\beta' = 0.707$), pairs of EPs appear. These edge modes are robust against disorder and thus exceptional edge modes robustly appear even with trivial bulk topology.

**Effective edge Hamiltonian and robustness of exceptional edge modes**. Robustness against the perturbation and the disorder is an important feature of topological edge modes. We note that conventional topological edge modes are fragile under the perturbations breaking the symmetry. To see what types of perturbations can sustain stable exceptional edge modes, we introduce a general one-dimensional effective Hamiltonian parametrized by wavenumber $k_y$,

$$H_{\text{edge}}(k_y) = \begin{pmatrix} E_0 + k_y + \alpha & i\beta + \gamma \\ i\beta - \gamma & E_0 - k_y - \alpha \end{pmatrix}, \qquad (3)$$

which describes the generic behavior of the low-energy dispersion of edge modes. The diagonal elements represent the linear dispersion of two edge modes without couplings, and the off-diagonal parts represent the non-Hermitian coupling. The case of $\alpha = \gamma = \text{Im } \beta = 0$ represents exceptional edge modes in the disorder-free system (Re $\beta \neq 0$ is necessary to generate EPs in exceptional edge modes). The effective Hamiltonian has the eigenenergy $E^{\pm}(k_y) = E_0 \pm \sqrt{(k_y + \alpha)^2 - \beta^2 - \gamma^2}$ and EPs at $k_y = -\alpha \pm \sqrt{\beta^2 + \gamma^2}$. The topological index associated with EPs can guarantee their presence in the complex wavenumber space in this case (see Supplementary Note 3).

As we consider the bulk gaps for the real parts of eigenenergies, the gapless edge modes remain when there exists a real wavenumber $k_y$ that satisfies Re $\sqrt{(k_y + \alpha)^2 - \beta^2 - \gamma^2} = 0$. Thus, we can conclude that $|\text{Im}\sqrt{\beta^2 + \gamma^2}| \leq |\text{Im } \alpha|$ is a necessary and sufficient condition to realize robust edge modes (see Methods).

Meanwhile, the exceptional edge modes remain when $-\alpha \pm \sqrt{\beta^2 + \gamma^2}$ is real, i.e., (i) Im $\alpha = 0$, (ii) Im $(\beta^2 + \gamma^2) = 0$, and (iii) Re $(\beta^2 + \gamma^2) > 0$. From conditions (ii), (iii), we can derive Im$\sqrt{\beta^2 + \gamma^2} = 0$ and thus can confirm that the condition for gapless edge modes must be satisfied under the condition for exceptional edge modes. We note that conditions (i), (ii) are equivalent to the condition for pseudo-Hermiticity[61], which ensures that the eigenenergies are either real or pairs of complex conjugate values (i.e., $(k_y + \alpha)^2 - \beta^2 - \gamma^2$ is real in the present case). In general, disorder in the existing terms satisfies this condition. Meanwhile, nonzero Re $\alpha$ breaks the time-reversal symmetry defined as $T = \sigma_y$, $TH(k_y)T^{-1} = H^*(-k_y)$. However, perturbation to Re $\alpha$ does not affect the stability of exceptional edge modes even if they accompany nontrivial bulk topology of a time-reversal-symmetric system.

We can also relate the robustness of the exceptional edge modes to the symmetry and the topology of EPs. In one-dimensional systems, EPs can robustly exist under the $PT$ symmetry, the $CP$ symmetry, the pseudo-Hermiticity, or the chiral symmetry[41–43]. In the case of $E_0 = \alpha = \gamma = \text{Im } \beta = 0$, the effective edge Hamiltonian exhibits the $PT$ symmetry $PT = \sigma_z$, $PTH(k_y)(PT)^{-1} = H^*(k_y)$, the $CP$ symmetry $CP = \sigma_x$, $CPH(k_y)(CP)^{-1} = -H^*(k_y)$, the pseudo-Hermiticity $\eta = \sigma_z$, $\eta H(k_y)\eta^{-1} = H^\dagger(k_y)$, and the chiral symmetry $\Gamma = \sigma_x$, $\Gamma H(k_y)\Gamma^{-1} = -H^\dagger(k_y)$. We note that the pseudo-Hermiticity here is in a narrower class than that considered in the previous paragraph, that is, the operator $\eta$ is restricted to a local operator that only acts on the inner degrees of freedom. To preserve the $PT$ symmetry and/or the chiral symmetry, we need Im $\alpha = \text{Im } \beta = \text{Re } \gamma = 0$. We can

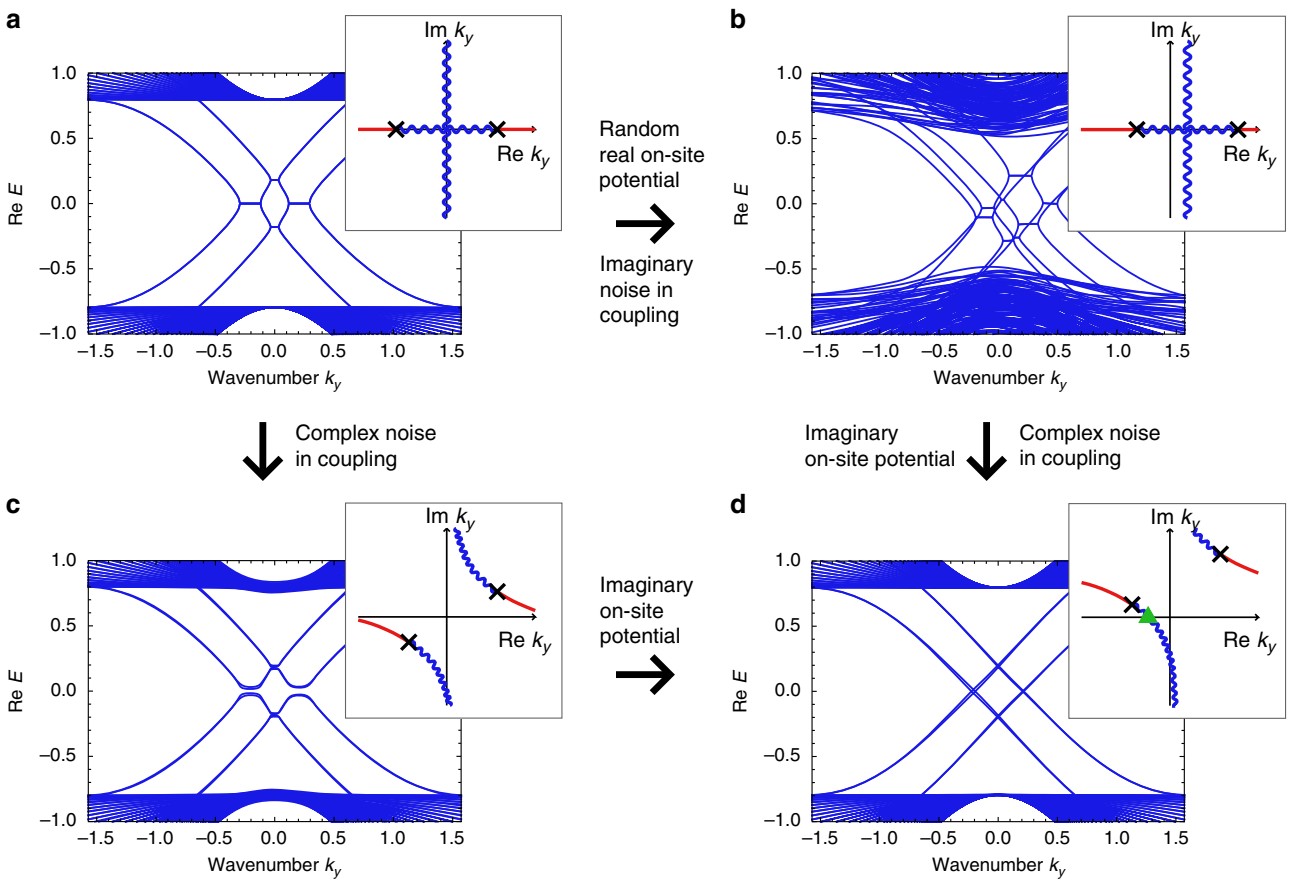

**Fig. 3 Edge modes in disordered systems and their robustness. a–d** Each inset shows the EPs (black crosses) and the curve of the degeneracy of the real (blue wave curves) and imaginary (red curves) parts of the edge eigenenergies in the complex wavenumber plane. Each inset corresponds to all the pairs of the edge modes with positive and negative group velocities. Because of the periodicity in the $y$ direction, the band structure corresponds to the behavior on the Re $k_y$ axis in the insets, whereas the EPs and the degeneracy curves in the complex wavenumber space are useful for predicting the behavior of the edge modes. **a** The main panel shows the obtained exceptional edge modes in the system without disorder. The parameters used are $u = -1$, $c = 0.2$, $\beta = 0.14$, $\beta' = 0.06$, and $\gamma = 0.05$. **b** The main panel shows the edge dispersion with on-site random real potentials and imaginary disorder in the coupling term. There still exist EPs and edge modes in the bulk energy gap. The noise widths are set to be $W = 0.5$ ($W = 0.02$) for the random real on-site potential (the imaginary noise in the non-Hermitian coupling). **c** The gap is opened and the edge modes no longer exist in the system with random Hermitian couplings. The noise width is set to be $W = 0.1$. **d** When we add imaginary on-site potentials, the edge modes are recovered even under the random Hermitian couplings. This is because the wavenumber should be real due to the periodic boundary condition in the $y$ direction and the real axis of the wavenumber plane crosses the degeneracy curve for the real parts of the eigenenergies (cf. the green triangle in the inset). However, EPs disappear from the edge modes. The noise width is the same as in panel **c** and the strength of the on-site imaginary potential is $g = 0.2$.

also confirm that the preservation of the $CP$ symmetry and/or the pseudo-Hermiticity requires Im $\alpha$ = Im $\beta$ = Im $\gamma$ = 0. From these equations, we can derive conditions (i), (ii) for realizing the exceptional edge modes discussed above and thus confirm that the exceptional edge modes robustly exist under sufficiently small $|$Im $\gamma|$ and one of the following symmetry: the $PT$ symmetry, the $CP$ symmetry, the chiral symmetry, or the pseudo-Hermiticity. If we increase $|$Im $\gamma|$, two EPs coalesce at critical strength of $|$Im $\gamma|$, and the exceptional edge modes disappear under larger $|$Im $\gamma|$. We can expect that the symmetry in the effective edge Hamiltonian is the same as that in the bulk and thus can utilize the symmetry as the guiding principle to predict what types of disorders remain exceptional edge modes.

To confirm the robustness of the exceptional edge modes in our model, we calculate the band structure with adding disorder (see Fig. 3). We show that the exceptional edge modes still exist robustly under certain types of disorders, i.e., the random real on-site potential and the imaginary noise in the coupling terms (see Methods for details). These disorders preserve the modified $PT$

symmetry $P'TH(k_x, k_y)(P'T)^{-1} \equiv H^*(-k_x, k_y)$ that has the same role as the $PT$ symmetry in the edge band structure. Thus, the result is consistent with the discussion in the previous paragraph. Also, the on-site non-Hermitian term, $ig\sigma_z \otimes I \otimes I$, recovers the robustness of the edge modes against the real noise in the coupling terms, which lifts the degeneracy in the edge bands without on-site terms. With the on-site non-Hermitian term, as the two edge modes avoid each other in the imaginary part of the energy, they are not degenerate and thus are prohibited to open the real gaps (similar feature has been observed in the previous study[10] at the interface between gain and loss regions). This avoidance protects the edge modes from opening gaps as understood from the perturbation theory (see Supplementary Note 4 for details). These results are consistent with the analysis of the effective edge Hamiltonian. In Supplementary Notes 11 and 12, we further discuss the symmetry of the disordered Hamiltonian and clarify its relation to the robustness of the exceptional edge modes in both topologically trivial and nontrivial systems. Especially, we demonstrate the existence of

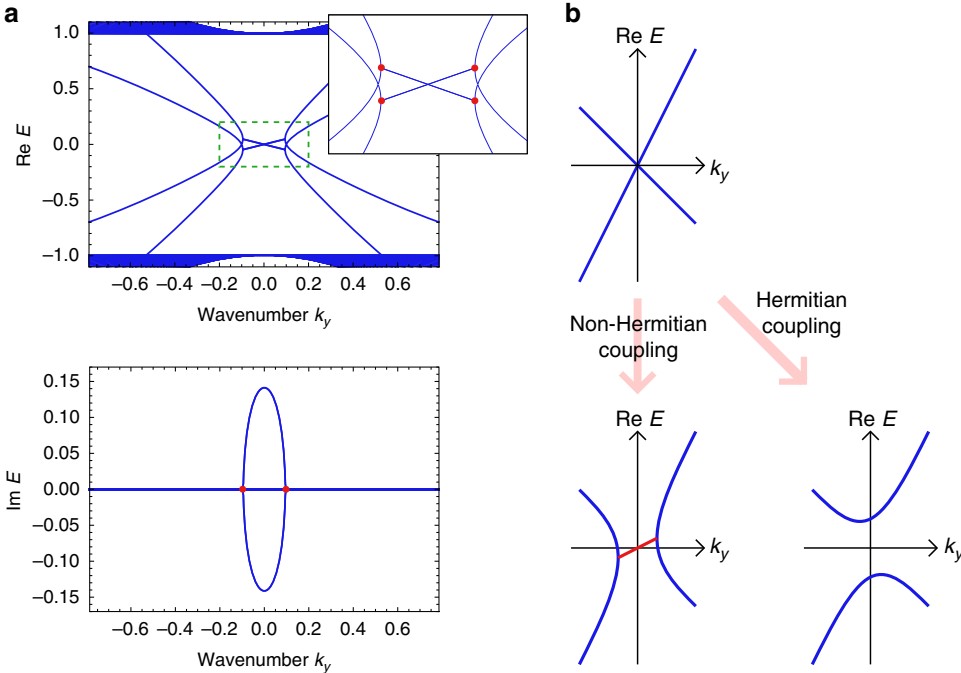

**Fig. 4 Band structure for realizing lasing edge modes with nonzero group velocities. a** Combining the Qi-Wu-Zhang (QWZ) model with large hoppings and the time-reversal QWZ model by a non-Hermitian coupling, we obtain the model with the edge band structure shown in the main panels. The edge dispersions between the pairs of EPs (red points) exhibit the nonzero imaginary part of the energy and the nonzero slope of the real part of the energy. Since the slope of the real part of the energy corresponds to the group velocity of the edge mode, these lasing edge modes have nonzero group velocity and propagate along the edge of the sample. The inset presents the enlarged view of the low momentum region indicated by the green dashed box and the red points represent the EPs in the edge dispersions. The parameters used are $u = -1$, $\beta = 0.2$, and $\beta' = 0.1$. **b** To obtain the lasing edge modes with nonzero group velocity, we utilize two edge modes which have opposite signs and different absolute values of the slope of the dispersion relation. By combining these edge modes by a non-Hermitian coupling term, we obtain exceptional edge modes, which can be applied to construct a topological insulator laser while Hermitian couplings open the gap owing to the avoided crossing.

the chiral-symmetry-protected exceptional edge modes and the importance of the modification of the $PT$ symmetry and the $CP$ symmetry for the protection of the exceptional edge modes.

In general, in Hermitian systems, the physical significance of the periodic table obtained from the bulk band topology is guaranteed by the bulk-edge correspondence that consistently predicts the presence or absence of robust gapless edge modes at open boundaries[4]. In contrast, in non-Hermitian cases, our findings force us to fundamentally alter this point of view. In particular, when $g \neq 0$ and $\gamma = 0$ in our model, there exist the robust gapless edge modes as in Fig. 3d ($g \neq 0$ and $\gamma \neq 0$), whereas the bulk topological invariant is trivial as inferred from the topological classification[23,30,31] (see Supplementary Note 10 for details). In other words, the robust gapless edge modes found here violate the bulk-edge correspondence and cannot be captured by the existing periodic tables[23,30,31] of non-Hermitian topological phases, thus challenging the conventional classification based on Bloch Hamiltonians.

**Application to amplifying edge modes.** Next, we show that amplified exceptional edge modes with nonzero group velocity can be realized. Specifically, we find that the general form of effective edge Hamiltonians is given by

$$H_{\text{edge}} = \begin{pmatrix} E_0 - ia\partial_y & i\beta \\ i\beta' & E_0 + i\partial_y \end{pmatrix}, \quad a \neq 1, \; \beta\beta' > 0. \quad (4)$$

We derive the dispersion relation of this effective Hamiltonian, $E(k_y) = E_0 + [(a-1)k_y \pm \sqrt{(a+1)^2 k_y^2 - 4\beta\beta'}]/2$, which exhibits EPs at $k_y = \pm 2\sqrt{\beta\beta'}/(a+1)$ and nonzero group velocity.

Although this Hamiltonian describes the generic behavior of lasing edge modes utilizing exceptional edge modes, we construct a concrete tight-binding model represented by the following Hamiltonian:

$$H = \begin{pmatrix} 2H_{\text{QWZ}} & i\beta\sigma_x \\ i\beta'\sigma_x & H_{\text{QWZ}}^* \end{pmatrix}, \quad (5)$$

where $H_{\text{QWZ}}$ is the Hamiltonian of the QWZ model. Figure 4a shows the edge band structure of this system. Nonzero imaginary parts of the eigenenergies appear only in the edge modes as in our first model. Also, the edge modes exhibit nonzero slopes of the real energy dispersion $\partial \text{Re} E/\partial k$, which correspond to nonzero group velocities. Thus, we can observe the amplified wave packet propagating along the edge of the sample, which allows us to stably transfer the energy and thus may find potential applications. We note that this Hamiltonian is neither time-reversal symmetric nor pseudo-Hermitian. The sum of the Chern numbers for the bands under the energy gap is zero in our model, which indicates the bulk triviality in the conventional sense. Therefore, the edge modes are protected not by the bulk band topology but by the EPs.

In general, exceptional edge modes are essential for this construction of a topological insulator laser. To obtain lasing edge modes, we must utilize a pair of edge modes localized at the same side whose dispersion relations cross each other without coupling terms. Also, to accomplish nonzero group velocity, the absolute values of the slopes of the edge energy bands must be different. Therefore, the degeneracy is not protected by the bulk band topology or the symmetry and thus can be resolved by Hermitian couplings as shown in Fig. 4b. On the contrary, non-Hermitian

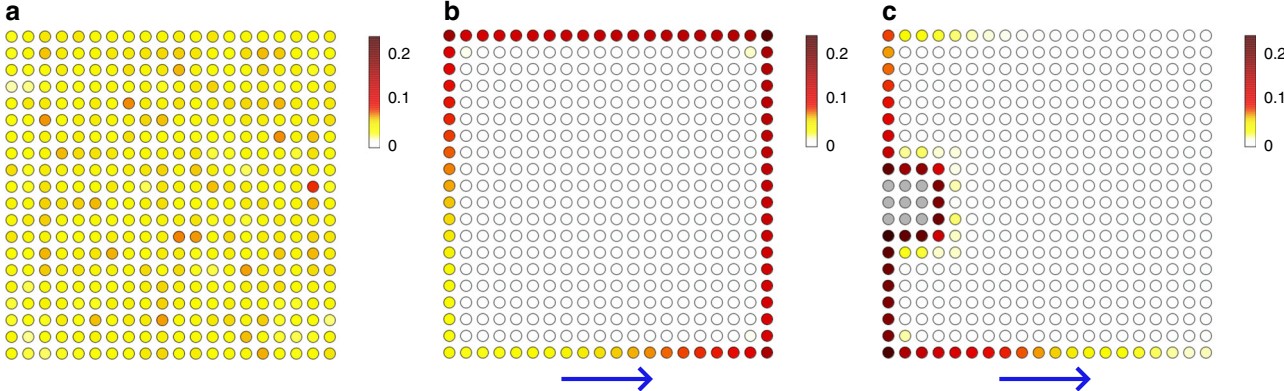

**Fig. 5 Real-space simulation of the lasing edge modes. a–c** The color represents the probability amplitude at each site, which is normalized so that the sum of squares gives unity. We set the parameter $u = -1$ throughout the calculations in this figure. **a** Without the non-Hermiticity ($\beta = \beta' = 0$), all the bulk modes can survive for a long time and thus lasing of edge modes fails to happen. **b** With the non-Hermitian coupling ($\beta = 0.2$, $\beta' = 0.1$), only the edge mode is enhanced even if we start from a random initial state. We can also confirm the propagation of the wave packet. **c** Starting from the excitation of one site at the edge, we confirm that a wave packet propagates along the edge of the sample. Even in the presence of distortion at the edge (represented by the gray sites), the edge wave packet avoids it and propagates without dissipation. The parameters used are $\beta = 0.9$ and $\beta' = 0.8$.

couplings lead to both the enhancement and the robustness of the edge mode. Thus, the lasing edge mode must be an exceptional edge mode.

We demonstrate the real-space dynamics of our topological insulator laser (Eq. (5)) by numerical calculations (see Supplementary movies 1-3). Figure 5 shows the snapshots for the real-space distributions of the probability densities of the wave functions. Without non-Hermitian coupling, $\beta = \beta' = 0$, the bulk oscillation survives. With non-Hermitian coupling, $\beta, \beta' \neq 0$, the bulk oscillation becomes much smaller than the edge oscillation in a short time, and only the edge mode remains even if we start with the random initial state. Also, we can confirm that the edge mode has nonzero group velocity. Furthermore, we introduce disorder on the edge and excite only one edge site. Then, we obtain the propagating edge mode without backscattering. This implies the robustness of the exceptional edge mode against the disorder at the edge. In contrast to the previous research[33,34], we do not need to introduce judicious gain along the edge. This difference can potentially facilitate the realization of topological insulator laser in various physical setups.

**Active matter realization of exceptional edge modes**. Analogous to the conventional topological edge modes[62], the exceptional edge modes can also exist in continuum systems. We construct a continuum toy model and confirm the existence of exceptional edge modes by calculating the band structure (see Methods and Supplementary Note 7 for the detail of the model). Figure 6 represents the edge band structure of the continuum model. Although the bulk bands are topologically trivial as in the tight-binding model (Eq. (1)), it exhibits the robust exceptional edge modes.

To show that exceptional edge modes are indeed realizable in realistic systems, we focus on a continuum active matter model. We consider chiral active matter without the left-right symmetry in which each particle moves on a clockwise (or counter-clockwise) circular trajectory (Fig. 7a). We mix clockwise and counterclockwise moving particles. We also assume that the chirality of active particles flips occasionally and the flipping rate $\gamma$ is symmetric between clockwise and counterclockwise moving particles. The active particles have long and narrow shapes. We assume that polar interaction acts on them, which aligns the neighboring particles and effectively appears in some self-propelled rods[63]. Anti-polar interaction is also allowed to exist

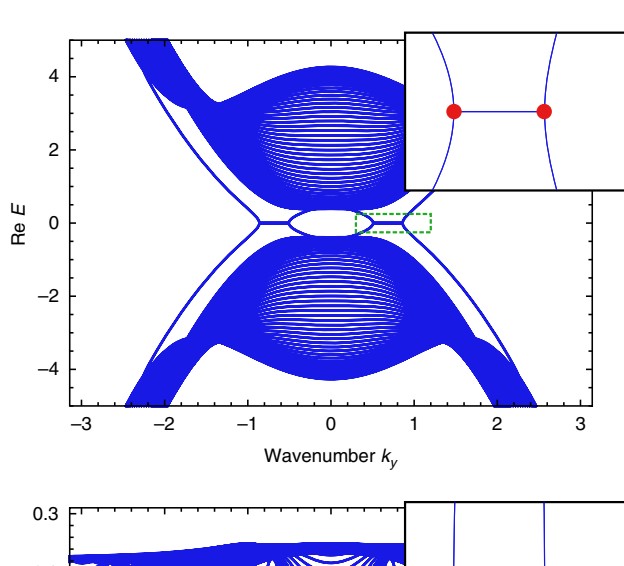

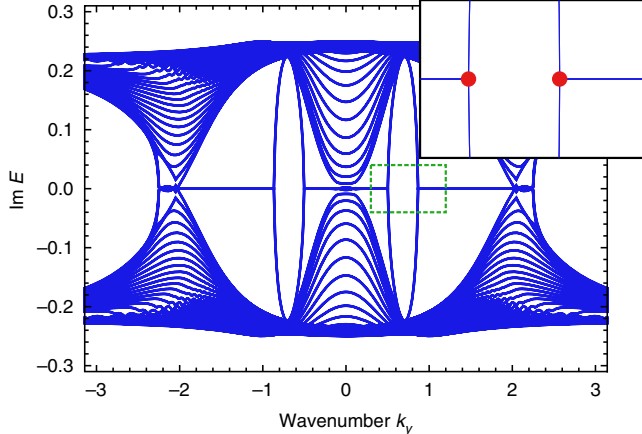

**Fig. 6 Exceptional edge modes in the continuum model.** The model is constructed from the continuum system with the Chern number 2 and its time reversal combined by a non-Hermitian coupling. We discretize the space and the equation, and numerically calculate the band structure under the open boundary condition in the $x$ direction and the periodic boundary condition in the $y$ direction. Although the model has topologically trivial bulk, it exhibits four edge bands per edge (doubly degenerated) protected by the exceptional points around Re $E = 0$ (indicated by the red points in the inset). The parameter used here are $M = 0.5$, $\beta = 0.5$, $a = 1$, $b = 0.3$, and $b' = 0.2$.

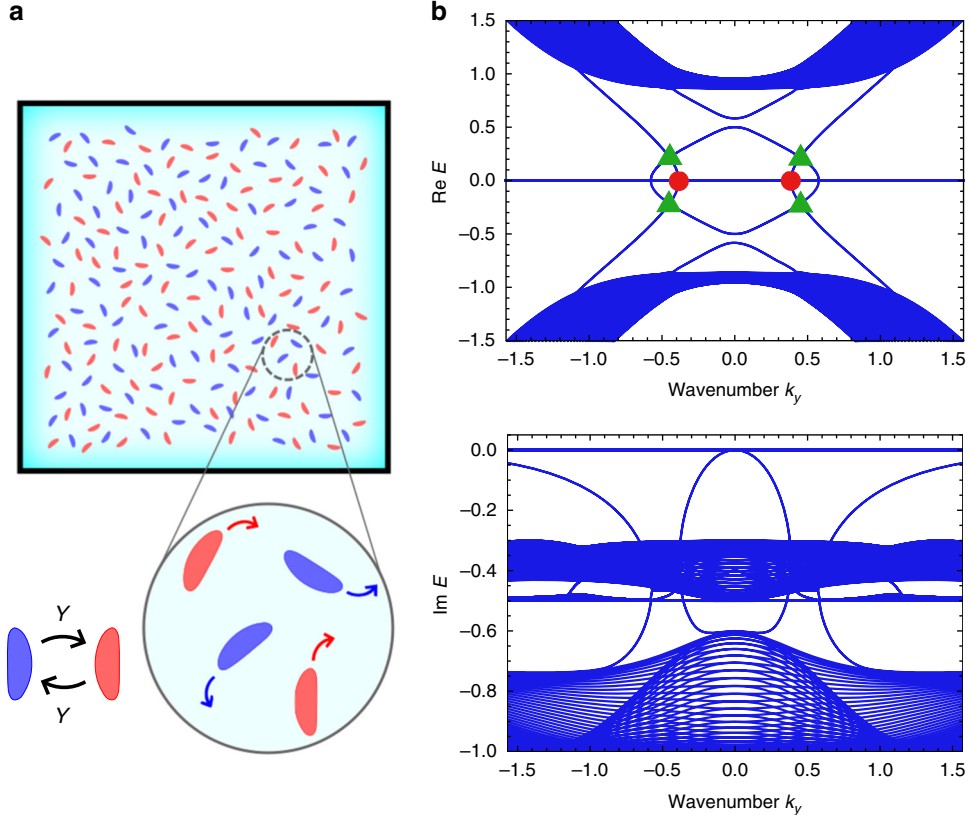

**Fig. 7 Exceptional edge modes in chiral active matter. a** Two-component chiral active matter is considered, where each active particle tends to move in the left (right) direction as shown in the blue (red) arrow. These two types of chiral active matter are mixed and the chirality can flip at a constant rate. **b** The edge band structure calculated for the cylindrical system is shown. The negativity of the imaginary part of eigenenergies implies the linear stability of the steady state. Since there exist four bands per edge in the bulk gap, the bulk topology is trivial in the conventional sense. The red circles represent the EPs that lead to the robustness of the edge modes. Also, we can find apparent edge-band crossings in the bulk real energy gap, which are denoted as the green triangles. However, the edge modes do not really cross there, because they avoid each other in their imaginary parts of the eigenenergies. Therefore, the apparent degeneracies are artifacts, and thus the gapless edge modes are not broken by disorder. The other crossings on the Re $E = 0$ axis correspond to the points where the edge bands come out of the bulk bands. The parameters used are $\omega_0 = 1$, $\nu^o = 0.5$, $\gamma = 0.3$, and $\beta = 0.5$.

between particles with opposite chirality. This setup can possibly be experimentally realized by utilizing bacteria[53], artificial L-shaped particles[54], or robotic rotors[52]. To be concrete, we expect that exceptional edge modes can appear in bacteria swimming between the two plates at the distance shorter than the bacteria length and in L-shaped active particles that are occasionally turned over (see Supplementary Note 8 for further details). Here, the crucial requirements for the experimental realization of exceptional edge modes are the flippable chirality and the momentum coupling.

In Fig. 7b, we show the existence of exceptional edge modes by numerically diagonalizing the effective Hamiltonian of our active matter model, which is derived by linearizing the hydrodynamic equations[46,64] (see Methods and Supplementary Methods). We confirm that a pair of EPs appear at the frequency $\omega = 0$ and support the robustness of the edge modes. Meanwhile, at a glance, there are degeneracies in the bulk gap. However, the edge modes avoid each other in the imaginary part of the frequency as in Fig. 3d, and thus these apparent degeneracies are robust against the disorder. We obtain two other crossings on the Re $E = 0$ axis, which correspond to the points where unprotected edge bands appear from the bulk bands around the axis (see Supplementary Note 9 for the detail on the function of these crossing). In realistic experimental situations, we expect that the oscillation of the fluctuation of the density or the velocity field propagates at the edge of the sample in the direction depending on the chirality of particles (i.e., clockwise or counterclockwise) when we apply the

perturbation with a small frequency compared to the bulk bandgap, which is almost equal to the frequency of rotation $\omega$. The imaginary parts of the eigenvalues are all nonpositive, and thus we need further modification of the active system to apply the proposed setup to lasing devices.

## Discussion

We revealed the existence of robust gapless edge modes unique to non-Hermitian systems by utilizing EPs. These edge modes, which we called exceptional edge modes, can exist even when the bulk topology is trivial. We also analyzed and confirmed the robustness of the edge modes by constructing the effective edge Hamiltonian. By utilizing these edge modes, we proposed a topological insulator laser whose edge modes were amplified and propagate along the edge. We also showed that the chiral active particles with chirality flipping can exhibit the exceptional edge modes and thus they can be realized in the upcoming experimental techniques of active matter, whereas the model analyzed here has only nonpositive imaginary parts of the eigenfrequencies and thus exhibits no lasing behavior.

The edge modes found here provide an alternative design principle to realize scattering-free edge current intrinsic to non-Hermitian systems, which is not based on the bulk topology and thus indicates that the conventional arguments on bulk topology, including the periodic tables[23,30,31], are insufficient to predict the presence or absence of robust edge modes in non-Hermitian

systems. Exceptional edge modes in higher-dimensional systems are important to further elucidate a nontrivial role of open boundaries in non-Hermitian systems. Furthermore, our active matter model demonstrates that hydrodynamics of active matter can be applied to non-Hermitian topological phenomena, indicating that active matter provides a useful platform for exploring non-Hermitian topology.

## Methods

**QWZ model and numerical calculations of the tight-binding model.** The Hamiltonian of the QWZ model in the wavenumber space can be described as $H(\mathbf{k}) = \sin k_x \sigma_x + \sin k_y \sigma_y + (u + \cos k_x + \cos k_y)\sigma_z$[59]. This model exhibits the two bulk bands with the Chern number $C = \pm 1$ and a bandgap around $E = 0$. By transforming it to the real-space basis via the inverse Fourier transformation, we obtain the Hamiltonian in the real space (see Supplementary Methods for the detailed description). To calculate the edge band structures, we transform only $\sin k_x$ and $\cos k_x$, and obtain the Hamiltonian of our model with the hybrid boundary conditions, i.e., the open boundary conditions in the $x$ direction and the periodic boundary conditions in the $y$ direction. We arrange the 50 unit cells in the $x$ direction and consider the $1 \times 50$ super-ribbon structures. Diagonalizing the obtained Hamiltonians, we calculate the edge band structures of our model.

We also introduce the disorder terms in Fig. 3. We use $a(x)\{I_2, \sigma_z\} \otimes \{I_2, \sigma_z\} \otimes \{I_2, \sigma_z\}$ and $b(x)\{\sigma_x, i\sigma_y\} \otimes \{I_2, \sigma_z\} \otimes \{\sigma_x, i\sigma_y\}$ as the random real on-site potential and the imaginary and real noise in the non-Hermitian coupling for each, where brackets mean that we introduce all the combinations made by choosing either one in each bracket. $a(x)$ and $b(x)$ are random values for each $x$ from uniform distributions ranging $[-W, W] \in \mathbb{R}$ or $i[-W, W] \in i\mathbb{R}$. We set $a(x)$ to be real and $W = 0.5$ in Fig. 3b. Also, we set $b(x)$ to be imaginary (real) for the imaginary (real) noise in the non-Hermitian coupling and $W = 0.02$ ($W = 0.1$) in Fig. 3b (Fig. 3c, d). We also consider on-site imaginary potential $ig\sigma_z \otimes I_2 \otimes I_2$ and set $g = 0.2$ in Fig. 3d.

To calculate the real-space dynamics of lasing modes in the finite system, we perform the inverse Fourier transformation for all the terms in $H(\mathbf{k})$ and impose the open boundary conditions both in the $x$ and $y$ directions. We arrange the $20 \times 20$ sites, which have four sublattices for each of the sites. We calculate the time evolution under the Hamiltonian by using the fourth-order Runge-Kutta method. We set the time step $dt = 0.001$ and use the parameters $u = -1$. The parameters of the non-Hermitian coupling are set to be $\beta = \beta' = 0$ in Fig. 5a, $\beta = 0.2$, $\beta' = 0.1$ in Fig. 5b, and $\beta = 0.9$, $\beta' = 0.8$ in Fig. 5c.

**Derivation of the condition for remaining gapless edge modes.** From the analysis of the effective edge Hamiltonian (Eq. (3)), we conclude that $|\mathrm{Im}\sqrt{\beta^2 + \gamma^2}| \le |\mathrm{Im}\,\alpha|$ is the necessary and sufficient condition for the existence of gapless edge modes. To show this, we start from the equation $\mathrm{Re}\sqrt{(k_y + \alpha)^2 - \beta^2 - \gamma^2} = 0$, which leads to the eigenenergy $E^\pm = E_0 \pm i\delta$ with $\delta$ being a real number. For a wavenumber satisfying this equation, the real parts of the two eigenenergies become the same, and thus we obtain the gapless edge modes. Then, we prove that the existence of such a wavenumber is equivalent to the condition $|\mathrm{Im}\sqrt{\beta^2 + \gamma^2}| \le |\mathrm{Im}\,\alpha|$. We consider $\alpha' = \mathrm{Im}\,\alpha$ and $k' = k_y + \mathrm{Re}\,\alpha$, and obtain $(k_y + \alpha)^2 = k'^2 - \alpha'^2 + 2ik'\alpha'$. Similarly, if we describe $\sqrt{\beta^2 + \gamma^2} = l + i\beta'$, we obtain $\beta^2 + \gamma^2 = l^2 - \beta'^2 + 2il\beta'$. To make $\sqrt{(k_y + \alpha)^2 - \beta^2 - \gamma^2}$ a real or a pure imaginary number, we have to set $\mathrm{Im}[(k_y + \alpha)^2 - \beta^2 - \gamma^2] = 0$ and thus consider $k' = l\beta'/\alpha'$. Then, $\mathrm{Re}[(k_y + \alpha)^2 - \beta^2 - \gamma^2] = (l^2 + \alpha'^2)(\beta'^2 - \alpha'^2)/\alpha'^2$ and this is zero or negative if and only if $\beta'^2 \le \alpha'^2$. Thus $|\mathrm{Im}\sqrt{\beta^2 + \gamma^2}| \le |\mathrm{Im}\,\alpha|$ is the necessary and sufficient condition for the existence of a wavenumber satisfying $\mathrm{Re}\sqrt{(k_y + \alpha)^2 - \beta^2 - \gamma^2} = 0$.

**Continuum model of the exceptional edge modes.** To construct the continuum model of the exceptional edge modes, we consider the $2 \times 2$ continuum Hamiltonian $H_{\mathrm{cont}}$ analyzed in the previous paper[62] (see Supplementary Note 7 for details), which describes the low-energy dispersion of a higher-Chern-number insulator (the Chern number $C = 2$). We combine it and its time-reversal counterpart $H_{\mathrm{cont}}^*$ by a non-Hermitian coupling. The obtained Hamiltonian is described as below:

$$H = \begin{pmatrix} M - \beta\nabla^2 & a(-i\partial_x - \partial_y)^2 & 0 & ib \\ a(-i\partial_x + \partial_y)^2 & -M + \beta\nabla^2 & ib' & 0 \\ 0 & ib & M - \beta\nabla^2 & a(i\partial_x - \partial_y)^2 \\ ib' & 0 & a(i\partial_x + \partial_y)^2 & -M + \beta\nabla^2 \end{pmatrix}, \quad (6)$$

where $M$, $\beta$, $a$, $b$, and $b'$ are real parameters. To numerically calculate the band structure, we must discretize the space and modify the derivative in the Hamiltonian into the difference between the neighboring sites. Here, we use central differences to discretize the first- and second-order $x$ derivatives, $\partial_x$ and $\partial_x^2$, in the Hamiltonian. We impose the open boundary condition in the $x$ direction. On the contrary, we consider the periodic boundary condition in the $y$ direction and convert the derivative $\partial_y$ into $ik_y$ to obtain the effective Bloch Hamiltonian. We discretize the space into 50 sites in the $x$ direction and set the discretization step to be unity. As in the tight-binding model, by numerically diagonalizing the obtained Hamiltonian, we calculate the edge band structure for the continuum model. We use the parameters $M = 0.5$, $a = 1$, $b = 0.5$, $b = 0.3$, and $b' = 0.2$ in the numerical calculation.

**Derivation of the effective Hamiltonian for two-component chiral active matter.** We start with the Vicsek-type model[65] with a constant rotational force on each chiral particle. We can derive hydrodynamic equations from such a particle model via the Boltzmann-Ginzburg-Landau approach[66,67]. First, we derive the Boltzmann equation of our particle model. Then, we consider the Fourier modes of the density function and only keep the leading order terms. The obtained equations can be interpreted as the hydrodynamic equations of active matter considered here. As a further step, we consider fluctuations of the density and the velocity fields from the unordered steady state. By linearizing the hydrodynamic equation with respect to those fluctuations, we finally obtain the linearized eigenequation. The coefficient matrix corresponds to the effective Hamiltonian of our model and is described as,

$$H = \begin{pmatrix} H_0 + A & C \\ -C^* & H_0^* + A \end{pmatrix}, \quad (7)$$

with $H_0$, $A$, and $C$ being

$$H_0 = \begin{pmatrix} 0 & -i\partial_x & -i\partial_y \\ -i\partial_x & 0 & -i(\omega_0 + \nu^o\Delta) \\ -i\partial_y & i(\omega_0 + \nu^o\Delta) & 0 \end{pmatrix}, \quad (8)$$

$$A = \begin{pmatrix} -i\gamma & 0 & 0 \\ 0 & -i\beta & 0 \\ 0 & 0 & -i\beta \end{pmatrix}, \quad (9)$$

$$C = \begin{pmatrix} i\gamma & 0 & 0 \\ 0 & i\beta & 0 \\ 0 & 0 & i\beta \end{pmatrix}. \quad (10)$$

Further details are provided in Supplementary Methods.

To numerically calculate the band structure, we must modify the derivative in our effective Hamiltonian into the difference between the neighboring sites in the discretized space as in the continuum model. Here, we use central differences to discretize the first- and second-order $x$ derivatives, $\partial_x$ and $\partial_x^2$, in our effective Hamiltonian. We impose the open boundary condition in the $x$ direction. On the contrary, we consider the periodic boundary condition in the $y$ direction and convert the derivative $\partial_y$ into $ik_y$ to obtain the effective Bloch Hamiltonian. We discretize the space into 50 sites in the $x$ direction and set the discretization step to be unity. As in the tight-binding model, by numerically diagonalizing the obtained Hamiltonian, we calculate the edge band structure of our active matter model. The parameters used in the numerical calculation are $\omega_0 = 1$, $\nu^o = 0.5$, $\gamma = 0.3$, and $\beta = 0.5$.

## Data availability
The data that support the plots within this paper and other findings of this study are available from the corresponding author on request.

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

## Acknowledgements
We thank Zongping Gong, Kohei Kawabata, Kyogo Kawaguchi, Daiki Nishiguchi, Shun Otsubo, Kazumasa Takeuchi, and Hiroki Yamaguchi for valuable discussions. K.S. is supported by World-leading Innovative Graduate Study Program for Materials Research, Industry, and Technology (MERIT-WINGS) of the University of Tokyo. Y.A. is supported by JSPS KAKENHI Grant Numbers JP16J03613 and JP19K23424. T.S. is supported by JSPS KAKENHI grant numbers JP16H02211 and JP19H05796. T.S. is also supported by Institute of AI and Beyond of the University of Tokyo.

## Author contributions
K.S., Y.A., and T.S. planned the project. K.S. performed the analytical and numerical calculations. K.S., Y.A., and T.S. analyzed and interpreted the results and wrote the manuscript.

## Competing interests
The authors declare no competing interests.
