## [Peer Review File · Nature Communications]

Reviewers' comments:

Reviewer #1 (Remarks to the Author):

In this manuscript, the authors couple a Qi-Wu-Zhang (QWZ) model with its time-reversal partner by non-hermitian coupling. The authors claim that in this coupled system, robust edge modes emerge, with gain and loss, which violates the bulk-edge correspondence as their bulk spectra are trivial. Based on this method, the authors propose a method to construct a topological laser using chiral active matter.

While finding some details interesting, I do not believe this work is of interest/importance to the general audience of Nature Communications.

Specifically, I have two main complaints: first, I think the claim of robust gapless edge states is misleading. As we can see from Fig. 2, these gapless edge states require a threshold of non-hermitian coupling. In other words, these edge modes with gain and loss can be removed by continuously tuning some parameters, so I do not see the robustness of these edge states. Accordingly, the authors' claim - a violation of the bulk-edge correspondence - is not convincing. In addition, I think there are some of the recent works, for example, Ref. 1, on the same topic. In Ref. 1., they achieved a thresholdless edge-gain state in a topological gap with non-hermitian coupling. So I think it's quite trivial to see a similar edge-gain effect in a trivial gap under some strong non-hermitian coupling.

On the application side, the authors propose a topological laser using the chiral active medium. It is unclear how feasible the proposal is without any detail about this chiral active medium.

As such, I do not think this manuscript warrants further consideration at Nature Communications and suggest transferring it to a more specific journal.

[1] Song, Alex Y., et al. " \mathcal{PT} -symmetric topological edge-gain effect." arXiv preprint arXiv:1910.10946 (2019).

Reviewer #2 (Remarks to the Author):

In this paper, the authors present a non-Hermitian extension of a Hermitian, topologically trivial model in two dimensions with time-reversal symmetry, which features edge modes that are protected by exceptional points. They show that these edge modes are robust against certain types of disorder, and find that they can appear in active matter with chiral properties.

Non-Hermitian topological phases are currently a very popular research subject, and are attracting much attention. To my knowledge, the appearance of edge states protected by exceptional points has not yet been discussed in the literature, and as such the work presented in this paper is relevant, new and timely. The scientific approach is sound, although I am not an expert on the Boltzmann-Ginzburg-Landau approach outlined in the supplementary information and am thus unable to judge the full

scientific merit of that part of the work. Overall I believe this contents of this paper form an important addition to the field and this paper deserves publication.

While the paper is generally well presented, I did find some parts of the text slightly misleading or confusing. I have specified this in more detail below along with additional questions and comments for the authors:

1. Throughout the text, the authors speak of a "bulk-edge correspondence", while I believe that it is more common to speak of a "bulk-boundary correspondence".
2. In the abstract, the authors write "[t]heir inherent eigenenergy spectra naturally lead to the application for amplifying edge modes that transfer the enhanced wave packets along the edge of the sample". I am not entirely sure what the authors mean to convey with this sentence, and it should probably be rewritten.
3. In the abstract as well as in the introduction, the authors write that the bulk-boundary correspondence in non-Hermitian models is broken. In my opinion the phrasing used in the introduction is too strong. More concretely, at the end of the second paragraph in the introduction, the authors write "the bulk-edge correspondence has been unestablished in the non-Hermitian case" (lines 24 and 25). This statement is not entirely true as there are works in which such a correspondence is established in non-Hermitian models, see, e.g., Refs. 25, 27 and 28 in the bibliography. Moreover, at the beginning of the third paragraph in the introduction, the authors write "[t]his indicates that the bulk-based classification cannot predict the existence or absence of edge modes in the non-Hermitian case" (lines 29 and 30). This is again not entirely true, as it has been shown in Refs. 23-25 and 27-31 that such a classification can be made. It is my opinion that in this manuscript it is shown that the bulk topology cannot conclusively predict the existence of boundary states in that it fails to predict the edge states with exceptional points while still accurately predicting the appearance of "ordinary" boundary states, i.e., those boundary states without exceptional points that are protected by the bulk topology. I believe this nuance should be made explicit.
4. In the third paragraph of the introduction, the authors mention that the exceptional edge modes that they find exhibit large imaginary parts of the eigenenergies thus making them suitable to realise topological lasers (lines 33 and 34). Firstly, I am wondering whether they mean large positive imaginary parts of the eigenenergies seeing that they probably want these states to be amplified and not decaying? Secondly, if they indeed mean that, they should mention explicitly in the text somewhere (introduction and/or conclusion), that the exceptional edge states appearing in the active matter only have negative imaginary energy and topological lasing thus does not take place in that context. Indeed, in the discussion, it is suggested implicitly that topological lasing also can be observed in active matter (see lines 221-223).
5. The sentence "The robustness of ... exceptional edge modes (lines 38 and 39) should be rephrased, e.g., "Our edge modes owe their robustness to the distinct..."
6. On page 3, the authors mention that "EPs join two edge dispersions like glue and make them robust against disorder" (lines 45 and 46). It is not discussed anywhere in the text why this makes them robust against disorder, and I am wondering whether this is due to the fact that this gluing of edge states corresponds to the appearance of a branch cut in the spectrum, which is also called a Fermi arc in the literature, see, e.g., Science 359, 1009 (2018), PRA 98, 042114 (2018), PRB 99, 041406 (2019)?
7. I believe it would be helpful to explicitly include the Bloch form of the QWZ model, which is now only mentioned in the methods section, somewhere on page 4.
8. In line 85, the authors write that their model has two layers of the QWZ model but from H it looks as if they have four layers of the QWZ model.
9. In lines 96 and 97, the authors write that "all the bulk modes can have zero imaginary parts of the eigenenergies". But in Fig. 1b, it is shown that all the bulk modes indeed do have zero imaginary part. Why do the authors say "can"?
10. At the top of page 5, the authors write "we consider the Hamiltonian with the additional Hermitian

coupling, H' ..." (lines 100 and 101). This should be rephrased to "with an additional" because now it sounds as if the whole of H' is the additional coupling, whereas this coupling is only represented by the second term in H' .

11. Does the term H' (line 101) preserve time-reversal symmetry? Or in other words, what does the additional Hermitian term do exactly?
12. The authors do not mention figure 2 anywhere in the main text.
13. How is Eq.(2) derived? Does it describe the edge states of the model H' or is it an unrelated model simply introduced to describe the generic behaviour of edge modes?
14. In lines 120 and 121, the authors write that "[t]he case of $\alpha = \gamma = \text{Im } \beta = 0$ represents exceptional edge modes in the disorder-free system". Why only $\text{Im } \beta = 0$ and not $\beta = 0$?
15. At the end of page 5, the authors write "[a]lso, the exceptional ..." (line 128). Is this an additional condition, i.e., that k_y should be real, next to the condition mentioned in line 127?
16. In lines 129, the authors say that "the first and the second conditions are equivalent to the condition for pseudo-Hermiticity". However, is the first condition alone, i.e., $\text{Im } \alpha = 0$, not already enough?
17. In lines 132 and 133 it says "nonzero $\text{Re } \alpha$ can break the time-reversal symmetry". Does it break time-reversal symmetry or not?
18. In the second paragraph on page 6 (lines 135-144), different types of disorder are discussed. Could the authors specify the different terms they add to the Hamiltonian specifically?
19. In lines 140-142 it says "since the two edge modes avoid each other in the imaginary part of the energy, they are not degenerate and thus are prohibited to open the real gaps". I do not see why this is the case. Could the authors perhaps explain?
20. I believe the sentence "[w]e next show that an edge mode with nonzero group velocity can be amplified by utilising exceptional edge modes" (lines 154 and 155) is a bit confusing, and does not really describe what the authors do in the subsequent text. I think it should probably be rephrased to something like "[n]ext we show that amplified exceptional edge modes with nonzero group velocity can be realised".
21. Is Eq. (3) derived from Eq. (4), or is it again an independent Hamiltonian (see comment 13)?
22. At the start of the discussion, the authors say that "[t]hese edge modes, which we called exceptional edge modes, can exist even when the bulk topology is trivial" (lines 218 and 219) thus suggesting that they can exist both when the bulk topology is trivial and non trivial. In the paper, however, the authors have only shown the existence of these edge modes in the case of trivial bulk topology. They should thus change this sentence.
23. In the methods, the authors mention they have used the fourth-order Runge-Kutta method. Could the authors give some details concerning the method in the supplementary information?
24. In the caption of Fig.1, the authors write that "[f]our gapless bands per edge exist in the bulk energy gap". This means that we should see eight gapless bands in the real part of the spectrum in 1(b), but instead we see four. Are the bands doubly degenerate?
25. At the end of the caption of Fig.1, the authors write "[t]hese EPs play the role of the 'glue' of the edge modes and thus prevent gap opening[s] by perturbations or disorders". To highlight this latter point, they could add to this sentence that this is shown in Fig.3.
26. In Fig.3, the insets of the figures show exceptional points. Firstly, which exceptional points are shown in this inset with regards to the main figures? Secondly, do these exceptional points still exist in Figs.3(c) and (d) because from the main figures it seems as if they have disappeared? Thirdly, could the authors perhaps better explain what they are showing in the insets and what it means (see also the next comment)?
27. At the end of the caption of Fig.3, it says "the edge modes are recovered even under the random Hermitian couplings because the real axis of the wavenumber plane crosses the degeneracy curve of the real parts of the eigenenergies". Is this the case because k_y should be real?
28. In the real part of the spectrum shown in Fig.6(b), several band crossings can be observed for the

in-gap states, of which two correspond to exceptional points and four to normal band crossings. However, there are two more crossings on the $\text{Re } E = 0$ axis. What do these correspond to? Also, if the four band crossings indicated by the green triangles are not really band crossings because they avoid each other in the imaginary parts of the energy, then what is the circular band exactly? Can it for example be pushed out of the gap into the bulk bands?

29. As a general comment to all the figures in the main text (Figs.1-6), in none of the caption of the figures it is mentioned explicitly for which parameters the plots are made. To relate the figures to the different terms discussed in the main text, it would be helpful to include this information.

30. In the supplementary information it is shown on page 2 why the bulk energy of the Hamiltonian in Eq. (1) in the main text has zero imaginary part by making use of the Bloch Hamiltonian. However, as bulk-boundary correspondence is broken, how can the authors be sure that results computed from the Bloch Hamiltonian have any merit when taking open boundary conditions?

31. Could the authors specify why Figs.4 and S1 are different as they seem to be plotted for the same model? Is it simply different parameter settings or are they plots for different models?

32. To which model in the main text does the continuum model in Eq.(S16) in the supplementary information correspond?

We are very grateful to Reviewer #1 for her/his careful reading of our manuscript and useful comments and criticism. We have very carefully studied all the comments and criticisms raised by the reviewer. Here, let us reply to each of them.

“In this manuscript, the authors couple a Qi-Wu-Zhang (QWZ) model with its time-reversal partner by non-hermitian coupling. The authors claim that in this coupled system, robust edge modes emerge, with gain and loss, which violates the bulk-edge correspondence as their bulk spectra are trivial. Based on this method, the authors propose a method to construct a topological laser using chiral active matter.

While finding some details interesting, I do not believe this work is of interest/importance to the general audience of Nature Communications.”

We appreciate the referee for pointing out that some important arguments have not been well-documented in the previous manuscript. In the following reply to each comment, we would like to explain why we believe that the present work is of great interest and importance to the general audience of Nature Communications.

“Specifically, I have two main complaints: first, I think the claim of robust gapless edge states is misleading. As we can see from Fig. 2, these gapless edge states require a threshold of non-hermitian coupling. In other words, these edge modes with gain and loss can be removed by continuously tuning some parameters, so I do not see the robustness of these edge states. Accordingly, the authors’ claim - a violation of the bulk-edge correspondence - is not convincing.”

We claim that exceptional edge modes are robust in the sense that a bandgap cannot be opened until the topology of the *edge* bands is altered. Specifically, we consider the branch-cut topology of the edge band structure around exceptional points. The topology of exceptional points can be changed only via their pair-annihilation or pair-creation, at which two or more exceptional points coalesce and topological invariants for each exceptional point (cf. Eq. (S13)) cannot be defined. Exceptional edge modes robustly exist until the parameters cross the singular point where the pair-annihilation or pair-creation of exceptional points occurs (corresponding to the parameters used in Fig. 2c). In addition, this view of the topology of exceptional points is natural in the sense that it is reminiscent of Hermitian gapless points such as Weyl points, which are also related to the band topology that can be altered only via the pair-annihilation or pair-creation (please see Refs. 41, 42 in the revised manuscript).

To further clarify the robustness of exceptional edge modes, we discuss the topological protection of exceptional points in one-dimensional systems like edge band structures. The protection of exceptional points in one-dimensional systems requires the symmetry, such as the pseudo-Hermiticity, the *PT* symmetry, the *CP* symmetry, or the chiral symmetry. We have reconsidered the symmetry of the effective edge Hamiltonian and found that either one of these symmetries can protect the exceptional edge modes. The effective edge Hamiltonian with zero base energy is described as

$$H_{\text{edge}}(k_y) = \begin{pmatrix} k_y + \alpha & i\beta + \gamma \\ i\beta - \gamma & k_y - \alpha \end{pmatrix},$$

and without the disorders $\alpha = \gamma = \text{Im } \beta = 0$, it exhibits the *PT* symmetry and the pseudo-Hermiticity $PT = \eta = \sigma_z$, $PTH(k_y)(PT)^{-1} = H^*(k_y)$, $PT(PT)^* = +1$, $\eta H(k_y)\eta^{-1} = H^\dagger(k_y)$, and the *CP* symmetry and the chiral symmetry $CP = \Gamma = \sigma_x$, $CPH(k_y)(CP)^{-1} = -H^*(k_y)$, $CP(CP)^* = +1$, $\Gamma H(k_y)\Gamma^{-1} = -H^\dagger(k_y)$. The preservation of the pseudo-Hermiticity or the *CP*

symmetry above leads to the following conditions: $\text{Im } \alpha = \text{Im } \beta = \text{Im } \gamma = 0$. We also obtain $\text{Im } \alpha = \text{Im } \beta = \text{Re } \gamma = 0$ as the condition for the preservation of the PT symmetry or the chiral symmetry. Under either of these conditions and small enough $|\text{Im } \gamma|$, the effective edge Hamiltonian satisfies the necessary and sufficient condition to support the exceptional edge modes derived in the main text. Therefore, under either one of these symmetries, we cannot remove the exceptional points unless $|\text{Im } \gamma|$ reaches the singular point where the exceptional points collide. In this way, we conclude that the topology and the symmetries protect the exceptional edge modes, which in turn guarantees their robustness as opposed to the referee's comment. We have added discussions and the new references noted above to mention these points as described in Summary of changes made (1) and (2).

“In addition, I think there are some of the recent works, for example, Ref. 1, on the same topic. In Ref. 1., they achieved a thresholdless edge-gain state in a topological gap with non-hermitian coupling. So I think it's quite trivial to see a similar edge-gain effect in a trivial gap under some strong non-hermitian coupling.”

First, we would like to note that our work has been done independently of the paper mentioned by the reviewer as Reference 1 [arXiv:1910.10946], which appeared on arXiv about two months before our manuscript. More importantly, we would like to emphasize that the main result of our work is very different from Reference 1 raised by the reviewer. In the present work, we propose and analyze the new mechanism to protect edge modes via the robustness of exceptional points through the analysis of the effective edge Hamiltonian and numerical calculations on disordered systems. While the paper mentioned by the reviewer and other previous researches (e.g. Ref. 31) discussed some aspects of exceptional points in the edge band structures, they have *not* discussed the crucial role of the exceptional points to stabilize the edge modes via the branch-cut topology, which is the main focus of our study. Thus, we firmly believe that the existence of these works does *not* diminish the novelty of our work.

In the present work, we propose the mechanism to protect gapless edge modes utilizing exceptional points, which is distinct from those discussed in previous research. Our proposed mechanism is quite general in the sense that the protection of exceptional points (and thus the proposed edge modes) only requires one of the following symmetries: the chiral symmetry, the pseudo-Hermiticity, the CP symmetry, and the PT symmetry (see also our reply above). Therefore, exceptional edge modes can appear in a broader class of systems than that considered in the paper mentioned by the reviewer (i.e., PT -symmetric systems). To further elucidate this point, we have conducted additional numerical calculations and confirmed that the chiral symmetry can protect the exceptional edge modes in the tight-binding model considered in the main text. The Hamiltonian of the model is

$$H(\mathbf{k}) = (u + \cos k_x + \cos k_y)I_2 \otimes I_2 \otimes \sigma_z + \sin k_y I_2 \otimes I_2 \otimes \sigma_y + \sin k_x \sigma_z \otimes I_2 \otimes \sigma_x \\ + cI_2 \otimes \sigma_x \otimes I_2 + i\frac{\beta + \beta'}{2}\sigma_x \otimes I_2 \otimes \sigma_x + i\frac{\beta - \beta'}{2}\sigma_x \otimes \sigma_z \otimes \sigma_x + \gamma\sigma_x \otimes \sigma_y \otimes \sigma_x,$$

and its chiral symmetry is defined as follows:

$$\Gamma = CT = \sigma_x \otimes \sigma_z \otimes \sigma_x, \quad \Gamma H(\mathbf{k})\Gamma^{-1} = -H^\dagger(\mathbf{k}), \quad \Gamma\Gamma^* = +1.$$

We have added a description of the other symmetries of this model in the Supplementary Information of the revised manuscript. If we add the imaginary noise in the non-Hermitian coupling considered in Fig. 3b and another disorder in the coupling $ib\sigma_x \otimes I_2 \otimes I_2$ (b is real and takes random values for each site), these terms break the PT symmetry, the CP symmetry, and the pseudo-Hermiticity and remain only the chiral symmetry. The figure below shows the result of the numerical calculation for the edge band structures of the non-Hermitian Bernevig-Hughes-Zhang model under these chiral-symmetry-preserving disorders. It is clear that the exceptional edge modes are robust even against the disorders. Furthermore, the eigenvalues do not appear as pairs of complex conjugates for each wavenumber k_y , which implies the absence of PT symmetry and other similar effects. Therefore, we conclude that the chiral symmetry can also protect the exceptional edge modes, and thus our proposal is applicable for a much wider class of systems than the previous research.

Additional Figure 1. Edge band structure under the chiral-symmetry-preserving disorder.

Regarding topological insulator lasers or edge-gain effect, we discuss the possibility of lasing edge modes with the nonzero group velocity. Such systems can exhibit scattering-free energy transfer and chiral motion of a lasing edge wave packet, which are characteristics of topological edge modes and can broaden the application of topological insulator laser. Therefore, together with the guiding principle for the protection of lasing edge modes, our proposal should have a direct impact on metamaterials designing. We have added discussions and a new reference in the main text and Supplementary Information to clarify these points as described in Summary of changes made (3), (4), and (S1).

“On the application side, the authors propose a topological laser using the chiral active medium. It is unclear how feasible the proposal is without any detail about this chiral active medium.

As such, I do not think this manuscript warrants further consideration at Nature Communications and suggest transferring it to a more specific journal.

The chiral active matter exhibiting exceptional edge modes can be realized by using bacteria, L-shaped particles, or robotic rotors that are utilized for experimental realization of chiral active matter in previous studies (cf. Refs. 49-51). Here, we provide a more detailed proposal for the experimental setup of the chiral active system. We consider bacteria swimming in the space sandwiched by upper and lower plates as shown in Figure below. We consider setting the distance between two plates slightly shorter than the bacteria length (about 1-2 μm , depending on the species of bacteria). Previous research (Ref. 50) has reported that, for example, *E. coli* shows the rotational movement near the surface. This is due to the rotation of flagella and their interaction to the surface, and thus the directions of rotation should be opposite for bacteria near the upper and lower surface. Therefore, this system exhibits the chirality and its flipping, which are the required elements for the active matter model of exceptional edge modes. The remained requirement is polar interaction. Self-propelled rods like bacteria can exhibit polar or nematic alignment interaction via the collisions or the hydrodynamic interaction, while the detailed mechanism of polar alignment is unclear. The previous research has suggested that the polar cluster tends to appear under strong repulsion and weak self-propulsion (see Ref. 61). Therefore, we may need to control the activity of bacteria by, for example, tuning the concentration of a solution. In short, we propose that the exceptional edge modes can be realized by bacteria (maybe weakly) swimming between upper and lower surfaces at the distance shorter than the bacteria length. We have added discussions in the main text and Supplementary Information to clarify these points as described in Summary of changes made (5) and (S2).

Additional Figure 2. Proposed experimental setup and the mechanism of the chirality and its flipping in bacterial motion. (a) Bacteria swim between upper and lower plates. (b) Bacteria near the upper (lower) surface show the counterclockwise (clockwise) motion due to the rotation of flagella. (c) The direction of rotation can be flipped occasionally by moving toward the other surface.

Once again, we are very grateful to Reviewer #1 for her/his valuable comments. In particular, we have revised the discussion about the protection of exceptional edge modes by the branch-cut topology of edge band structures. We do hope that the revised manuscript, together with our reply above, will meet with the reviewer’s approval.

We are very grateful to Reviewer #2 for her/his elaborate review and a positive appreciation of our work. We have very carefully studied all the comments and criticisms raised by the reviewer. Here, let us reply to each of them.

“In this paper, the authors present a non-Hermitian extension of a Hermitian, topologically trivial model in two dimensions with time-reversal symmetry, which features edge modes that are protected by exceptional points. They show that these edge modes are robust against certain types of disorder, and find that they can appear in active matter with chiral properties.

Non-Hermitian topological phases are currently a very popular research subject, and are attracting much attention. To my knowledge, the appearance of edge states protected by exceptional points has not yet been discussed in the literature, and as such the work presented in this paper is relevant, new and timely. The scientific approach is sound, although I am not an expert on the Boltzmann-Ginzburg-Landau approach outlined in the supplementary information and am thus unable to judge the full scientific merit of that part of the work. Overall I believe this contents of this paper form an important addition to the field and this paper deserves publication.

While the paper is generally well presented, I did find some parts of the text slightly misleading or confusing. I have specified this in more detail below along with additional questions and comments for the authors:”

We thank the reviewer for a positive appreciation of our results. We address each comment raised by the reviewer below.

“1. Throughout the text, the authors speak of a “bulk-edge correspondence”, while I believe that it is more common to speak of a “bulk-boundary correspondence”.”

In our understanding, the bulk-boundary correspondence has a wider meaning as it indicates the existence of localized modes at the boundary of bulk regions with *different* topological invariants. Since we focus on topological systems surrounded by vacuum, we would think that it is appropriate to say “the violation of the bulk-edge correspondence” in this paper.

“2. In the abstract, the authors write “[t]heir inherent eigenenergy spectra naturally lead to the application for amplifying edge modes that transfer the enhanced wave packets along the edge of the sample”. I am not entirely sure what the authors mean to convey with this sentence, and it should probably be rewritten.”

In this sentence, we intended to mean that the complex eigenenergy spectra are applicable for lasing edge modes that propagate along the edge of the sample. The propagation enables us to transfer the wave packet and its energy without back-scattering, which may find potential applications. We have revised the sentence to clarify this point as summarized in (6).

“3. In the abstract as well as in the introduction, the authors write that the bulk-boundary correspondence in non-Hermitian models is broken. In my opinion the phrasing used in the introduction is too strong. More concretely, at the end of the second paragraph in the introduction, the authors write “the bulk-edge correspondence has been unestablished in the non-Hermitian case” (lines 24 and 25). This statement is not entirely true as there are works in which such a correspondence is established in non-Hermitian models, see, e.g., Refs. 25, 27 and 28 in the bibliography. Moreover, at the beginning of the third paragraph in the introduction, the authors write “[t]his indicates that the bulk-based classification cannot predict the existence or absence of edge modes in the non-Hermitian case” (lines 29 and 30). This is again not entirely true, as it has been shown in Refs. 23-25 and 27-31 that such a classification can be made. It is my opinion that in this manuscript it is shown that the bulk topology cannot conclusively predict the existence of boundary states in that it fails to predict the edge states with exceptional points while still accurately predicting the appearance of “ordinary” boundary states, i.e., those boundary states without exceptional points that are protected by the bulk topology. I believe this nuance should be made explicit.”

We thank the reviewer for this insightful comment. In our understanding, the case where the bulk-boundary correspondence is established is restricted, e.g., to one-dimensional chiral-symmetric systems and two-dimensional systems without symmetries or with some symmetries that prohibit the non-Hermitian skin effect. Therefore, “the bulk-edge correspondence has been unestablished” may be too strong but it would be fair to say as “not fully established”. On the latter part of the comment, we find that the explanation of the exception in the prediction from the conventional bulk topology described in the sixth sentence of this comment is very clear. We thank the reviewer again for this valuable suggestion and have revised sentences to implement these nuances as summarized in (7).

“4. In the third paragraph of the introduction, the authors mention that the exceptional edge modes that they find exhibit large imaginary parts of the eigenenergies thus making them suitable to realise topological lasers (lines 33 and 34). Firstly, I am wondering whether they mean large positive imaginary parts of the eigenenergies seeing that they probably want these states to be amplified and not decaying? Secondly, if they indeed mean that, they should mention explicitly in the text somewhere (introduction and/or conclusion), that the exceptional edge states appearing in the active matter only have negative imaginary energy and topological lasing thus does not take place in that context. Indeed, in the discussion, it is suggested implicitly that topological lasing also can be observed in active matter (see lines 221-223).”

As the reviewer has mentioned, we need a large *positive* imaginary part of the eigenenergy for realizing a lasing mode. We may need tuning the global gain or loss to realize positive imaginary parts of the eigenenergies of exceptional edge modes. Thus, the active matter model analyzed in this paper does not exhibit lasing and requires further modifications if one intends to apply them to actual lasing devices. We have revised and added sentences to clarify this point as summarized in (8).

“5. The sentence “The robustness of ... exceptional edge modes (lines 38 and 39) should be rephrased, e.g., “Our edge modes owe their robustness to the distinct...””

We have implemented this change suggested by the reviewer in the revised manuscript as described in Summary of changes made (9).

“6. On page 3, the authors mention that “EPs join two edge dispersions like glue and make them robust against disorder” (lines 45 and 46). It is not discussed anywhere in the text why this makes them robust against disorder, and I am wondering whether this is due to the fact that this gluing of edge states corresponds to the appearance of a branch cut in the spectrum, which is also called a Fermi arc in the literature, see, e.g., Science 359, 1009 (2018), PRA 98, 042114 (2018), PRB 99, 041406 (2019)?”

We thank the reviewer for this interesting comment. The gluing by the exceptional points is closely related to the so-called bulk Fermi arc and other non-Hermitian nodal structures as mentioned by the reviewer. Especially, the robustness of exceptional points in the complex wavenumber space is supported by the branch-cut topology like in the Riemann surface structure in bulk bands, i.e., a bulk Fermi arc, which accompanies the curve in the wavenumber space where the real energy gap is closed. For the gluing of edge modes, we must consider the protection of exceptional points in 1-dimensional wavenumber space. This is made possible with the help of symmetries, such as the *PT* symmetry, the *CP* symmetry, the pseudo-Hermiticity, and the chiral symmetry (see, e.g., Ref. 40 in the revised manuscript). One-dimensional bands sandwiched by two topologically protected exceptional points show the same real part of the energy like in the exceptional edge mode, and thus the gluing can be realized by the exceptional points.

To further elucidate the above point, we confirmed that the topology and the symmetries protect the exceptional points in exceptional edge modes by analyzing the symmetry of the effective edge Hamiltonian. From the symmetry (the *PT* symmetry, the *CP* symmetry, the chiral symmetry, or the pseudo-Hermiticity) of the effective Hamiltonian and small enough $|\text{Im } \gamma|$, we can derive the necessary and sufficient condition for realizing the exceptional edge modes proposed in the main text. The exceptional points in the edge modes can be removed only by the pair-annihilation, at which the topological invariant for the exceptional points cannot be defined. Therefore, we conclude

that the topology and the symmetries protect the exceptional edge modes. The detailed discussion is described in the revised main text. We note that the conditions for realizing the exceptional edge modes are looser than the symmetry constraint above and lead to the broader class of the pseudo-Hermiticity, where η can be a global operator that satisfies $\eta H \eta^{-1} = H^\dagger$.

We have also reexamined the symmetry of our tight-binding model and confirmed the consistency with the discussion above. We checked that the two-layered non-Hermitian Bernevig-Hughes-Zhang model has the PT symmetry, the CP symmetry, the chiral symmetry, and the pseudo-Hermiticity (see Supplementary Information in the revised manuscript for details). We also found that the following modified PT symmetries, CP symmetry, and pseudo-Hermiticity can be defined in this model.

$$P'T = I_2 \otimes I_2 \otimes \sigma_z, \quad P'TH(k_x, k_y)(P'T)^{-1} = H^*(-k_x, k_y), \quad (PT \text{ symmetry})$$

$$CP' = \sigma_y \otimes \sigma_z \otimes \sigma_y, \quad CP'H(k_x, k_y)(CP')^{-1} = -H^*(-k_x, k_y), \quad (CP \text{ symmetry})$$

$$\eta' = \sigma_y \otimes I_2 \otimes I_2, \quad \eta'H(k_x, k_y)\eta'^{-1} = H^\dagger(-k_x, k_y), \quad (\text{pseudo-Hermiticity})$$

These modified symmetries can play a similar role to the conventional ones in the edge band when we impose the open boundary condition in the x -direction. The modified PT symmetry is preserved under the random real onsite potential and the imaginary noise in the non-Hermitian coupling considered in Fig. 3b. Complex noise in the non-Hermitian coupling in Fig. 3c breaks all the symmetries and thus opens a gap in the edge band structure. These are consistent with the discussion in the previous paragraph.

Additional Figure 3. Edge band structure under the chiral-symmetry-preserving disorder.

By conducting additional numerical calculations, we also confirm the robustness of exceptional edge modes against disorders that preserve the chiral symmetry and breaks the other symmetries.

We consider the additional Hermitian coupling $\alpha\sigma_x \otimes \sigma_y \otimes \sigma_x$, the imaginary noise in the non-Hermitian coupling considered in Fig. 3b, and another disorder term $ib\sigma_x \otimes I_2 \otimes I_2$ (α and b are real and b takes random values for each site). The figure above presents the result of the calculation of the edge bands under the disorder and shows that exceptional edge modes still exist under the disorder. One can check that all the topological invariants predicted from the periodic table (cf. Refs. 30, 31) are trivial (see revised Supplementary Information for details). Thus, the robust edge modes protected by the chiral symmetry clearly breaks the bulk-edge correspondence in non-Hermitian systems, while their protection is consistent with the protection of exceptional points by topology and symmetry.

On the other hand, we would like to note that the conventional PT symmetry, the conventional CP symmetry, and the modified pseudo-Hermiticity cannot contribute to the protection of the exceptional edge modes. One can check this from the analysis corresponding to Fig. 3d (or Supplementary Fig. 6 in the revised manuscript). The imaginary on-site potential considered in Fig. 3d, $ig\sigma_z \otimes I_2 \otimes I_2$, preserves the conventional PT symmetry PT , the conventional CP symmetry CP , and the modified pseudo-Hermiticity η' and breaks the other symmetries. If we add the imaginary on-site potential, the exceptional points disappear as can be seen from Fig. 3d, which implies that the conventional PT and CP symmetries and the modified pseudo-Hermiticity are independent of the existence of the exceptional edge modes. We can further discuss the irrelevance of these symmetries from the fact that the symmetry operators include the space inversion and thus convert an edge mode into another edge mode localized at the opposite side (see revised Supplementary Information for details). We have added discussions and new references to clarify this point as summarized in (10), (11), (S3), and (S4).

“7. I believe it would be helpful to explicitly include the Bloch form of the QWZ model, which is now only mentioned in the methods section, somewhere on page 4.”

We have implemented this change suggested by the reviewer in the revised manuscript as described in Summary of changes made (12).

“8. In line 85, the authors write that their model has two layers of the QWZ model but from H it looks as if they have four layers of the QWZ model.”

As mentioned by the reviewer, the model has four layers of the QWZ model. Two of them are the time-reversal counterparts of the others. We have corrected the manuscript as described in Summary of changes made (13).

“9. In lines 96 and 97, the authors write that “all the bulk modes can have zero imaginary parts of the eigenenergies”. But in Fig. 1b, it is shown that all the bulk modes indeed do have zero imaginary part. Why do the authors say “can”?”

In the sentence, we mention the general case where exceptional edge modes appear. In general, bulk modes can also have nonzero imaginary parts of energies (cf. in Figs. 6 and 7). This is why we use “can” in this sentence. We have revised the manuscript to clarify that we consider general cases in this sentence as summarized in (14).

“10. At the top of page 5, the authors write “we consider the Hamiltonian with the additional Hermitian coupling, H' ...” (lines 100 and 101). This should be rephrased to “with an additional” because now it sound

as if the whole of H' is the additional coupling, whereas this coupling is only represented by the second term in H' .”

We thank the reviewer for this useful comment. In response to the revision related to comment 22, we have revised this sentence and removed the words “with the additional Hermitian coupling”. Please see also the reply to comment 22.

“11. Does the term H' (line 101) preserve time-reversal symmetry? Or in other words, what does the additional Hermitian term do exactly?”

The Hamiltonian H' still has the time-reversal symmetry. Here, we can define the time-reversal operator as follows:

$$T = \sigma_y \otimes I_2 \otimes I_2, \quad TH'(\mathbf{k})T^{-1} = (H')^*(-\mathbf{k}), \quad TT^* = -1.$$

The additional term introduces the coupling preserving the time-reversal symmetry like the Rashba term in the model of the quantum spin Hall effect in Ref. 3 and thus can open a gap in the trivial insulator for the Hermitian case. We have also realized that this term breaks the pseudo-Hermiticity defined as

$$\eta = \sigma_z \otimes I_2 \otimes I_2, \quad \eta H(\mathbf{k})\eta^{-1} = H^\dagger(\mathbf{k}), \quad \eta\eta^* = +1.$$

We have added discussions to clarify this point as summarized in (15).

“12. The authors do not mention figure 2 anywhere in the main text.”

We thank the reviewer for this comment. We have added words to refer to Figure 2 and to clarify its relation to the main text as summarized in (16).

“13. How is Eq.(2) derived? Does it describe the edge states of the model H' or is it an unrelated model simply introduced to describe the generic behaviour of edge modes?”

Equation (2) ((3) in the revised manuscript) phenomenologically describes the low-energy dispersion of exceptional edge modes and the more generic behavior of edge modes. Thus, we have not derived the equation directly from the microscopic Hamiltonian H' , but it is introduced as an effective model to describe the behavior of edge modes as pointed out by the referee. We have revised the manuscript to clarify this point as summarized in (17).

“14. In lines 120 and 121, the authors write that “[t]he case of $\alpha = \gamma = \text{Im } \beta = 0$ represents exceptional edge modes in the disorder-free system”. Why only $\text{Im } \beta = 0$ and not $\beta = 0$?”

The real part of β represents the non-Hermitian coupling and is necessary for realizing the robust exceptional edge modes. If β becomes zero, the effective edge Hamiltonian exhibits linear dispersions $E = \pm k$ and thus has no robust exceptional points. We have added discussions to clarify this point as summarized in (18).

“15. At the end of page 5, the authors write “[a]lso, the exceptional ...” (line 128). Is this an additional condition, i.e., that k_y should be real, next to the condition mentioned in line 127?”

Gapless edge modes appear when there exists a real wavenumber k_y satisfying $\text{Re}\sqrt{(k_y + \alpha)^2 - \beta^2 - \gamma^2} = 0$. On the other hand, exceptional edge modes appear when there exists a real wavenumber k_y where both the real and imaginary parts of $\sqrt{(k_y + \alpha)^2 - \beta^2 - \gamma^2}$

become zero. The second and third conditions in line 128 (line 150 in the revised manuscript) lead to $\text{Im}\sqrt{\beta^2 + \gamma^2} = 0$. Therefore, the condition in line 127 (line 148 in the revised manuscript) $|\text{Im}\sqrt{\beta^2 + \gamma^2}| \leq |\text{Im } \alpha|$ is always satisfied under the conditions in line 128 (line 150 in the revised manuscript). We have added discussions to clarify this point as summarized in (19).

“16. In lines 129, the authors say that “the first and the second conditions are equivalent to the condition for pseudo-Hermiticity”. However, is the first condition alone, i.e., $\text{Im } \alpha = 0$, not already enough?”

If we allow $\text{Im}(\beta^2 + \gamma^2) \neq 0$, the eigenenergy becomes complex and does not appear as the pair of the complex conjugate, which implies the breakdown of the pseudo-Hermiticity. We can see this from the fact that $(k_y + \alpha)^2 - \beta^2 - \gamma^2$ becomes complex and thus $\sqrt{(k_y + \alpha)^2 - \beta^2 - \gamma^2}$ is neither real nor pure imaginary. Therefore, we need the second condition, $\text{Im}(\beta^2 + \gamma^2) = 0$, for ensuring pseudo-Hermiticity. We have added discussions to clarify this point as summarized in (20).

“17. In lines 132 and 133 it says “nonzero $\text{Re } \alpha$ can break the time-reversal symmetry”. Does it break time-reversal symmetry or not?”

A nonzero $\text{Re } \alpha$ breaks the time-reversal symmetry here. The effective edge Hamiltonian exhibits the time-reversal symmetry, $T = \sigma_y$, $TH(\mathbf{k})T^{-1} = H^*(-\mathbf{k})$, $TT^* = -1$, when there are no disorders, i.e., $\alpha = \gamma = \text{Im } \beta = 0$. $\text{Re } \alpha \neq 0$ leads to the time-reversal-symmetry-breaking term $\alpha\sigma_z$. We have removed “can” in this sentence and added discussion to clarify the argument as summarized in (21).

“18. In the second paragraph on page 6 (lines 135-144), different types of disorder are discussed. Could the authors specify the different terms they add to the Hamiltonian specifically?”

We introduce the following terms as the disorder:

$$a(x) \left\{ \begin{array}{c} I_2 \\ \sigma_z \end{array} \right\} \otimes \left\{ \begin{array}{c} I_2 \\ \sigma_z \end{array} \right\} \otimes \left\{ \begin{array}{c} I_2 \\ \sigma_z \end{array} \right\}, \text{ (random real on-site potential),}$$

$$b(x) \left\{ \begin{array}{c} \sigma_x \\ i\sigma_y \end{array} \right\} \otimes \left\{ \begin{array}{c} I_2 \\ \sigma_z \end{array} \right\} \otimes \left\{ \begin{array}{c} \sigma_x \\ i\sigma_y \end{array} \right\}, \text{ (imaginary and real noise in the non-Hermitian coupling),}$$

where brackets indicate that we introduce all the combinations made by choosing either one in each bracket. The parameters $a(x)$ and $b(x)$ take uniformly random values from $[-W, W] \in \mathbb{R}$ or $i[-W, W] \in i\mathbb{R}$ for each site x . $a(x)$ is restricted to real value and $b(x)$ is restricted to imaginary (real) value for the imaginary (real) noise in the non-Hermitian coupling. We set $W = 0.5$ for the random real on-site potential, $W = 0.02$ for the imaginary noise in the non-Hermitian coupling, and $W = 0.1$ for the imaginary noise in the non-Hermitian coupling. These disorders break the CP symmetry, the chiral symmetry, and the pseudo-Hermiticity but remain the modified

PT symmetry. We have added discussions to specify the disorder terms used in this analysis in the revised manuscript as described in Summary of changes made (22).

“19. In lines 140-142 it says “since the two edge modes avoid each other in the imaginary part of the energy, they are not degenerate and thus are prohibited to open the real gaps”. I do not see why this is the case. Could the authors perhaps explain?”

We can explain the reason for the prohibition of opening the real gaps by utilizing the perturbation theory. The main reason is that the real gaps can be opened only via the degeneracy of edge modes while adding small perturbation cannot make degeneracy. We consider two edge modes $|\psi_1(k_y)\rangle$, $|\psi_2(k_y)\rangle$ of the Hamiltonian $H(k_y)$ with the eigenvalues $E_1(k_y)$, $E_2(k_y)$. We assumed that the other eigenvalues are separated far from $E_1(k_y)$ and $E_2(k_y)$. The perturbation theory predicts that if the expected value of the perturbation ϵV is much smaller than $|E_1(k_y) - E_2(k_y)|$, the two eigenvalues of $H(k_y) + \epsilon V(k_y)$ corresponding to $E_1(k_y)$ and $E_2(k_y)$ are $E'_1(k_y) = E_1(k_y) + \mathcal{O}(\epsilon)$ and $E'_2(k_y) = E_2(k_y) + \mathcal{O}(\epsilon)$. Thus, the change of the distance of the two eigenvalues is the order of ϵ . Since we assume that ϵ is much smaller than the original distance of two eigenvalues $|E_1(k_y) - E_2(k_y)|$, the degeneracy is prohibited under the small perturbation. Without the degeneracy, $E'_1(k_y)$ and $E'_2(k_y)$ should be continuous for the change of k_y and ϵ , which implies that $E'_{1,2}(k_y)$ depicts the smooth curve connecting $E'_{1,2}(k_y = -\pi)$ and $E'_{1,2}(k_y = \pi)$. Since $\text{Re}E'_{1,2}(k_y = -\pi) > (<)0$ and $\text{Re}E'_{1,2}(k_y = \pi) < (>)0$ are satisfied as in the nonperturbed case, $E'_{1,2}(k_y)$ crosses the $E = 0$ axis i.e., these edge modes are gapless under the small perturbation. We have added discussions to clarify this point as summarized in (23).

“20. I believe the sentence “[w]e next show that an edge mode with nonzero group velocity can be amplified by utilising exceptional edge modes” (lines 154 and 155) is a bit confusing, and does not really describe what the authors do in the subsequent text. I think it should probably be rephrased to something like “[n]ext we show that amplified exceptional edge modes with nonzero group velocity can be realised”.

We thank the reviewer for this useful comment. We have implemented this change suggested by the reviewer in the revised manuscript as described in Summary of changes made (24).

“21. Is Eq. (3) derived from Eq. (4), or is it again an independent Hamiltonian (see comment 13)?”

Please see the reply to comment 13 for the detail. Equation (3) is not directly derived from Eq. (4). (The equations mentioned above correspond to Eqs. (4) and (5) for each in the revised manuscript.) We have revised a sentence to clarify this point as summarized in (25).

“22. At the start of the discussion, the authors say that “[t]hese edge modes, which we called exceptional edge modes, can exist even when the bulk topology is trivial” (lines 218 and 219) thus suggesting that they can exist both when the bulk topology is trivial and non trivial. In the paper, however, the authors have only shown the existence of this edge modes in the case of trivial bulk topology. They should thus change this sentence.”

We thank the reviewer for this important comment. While we concentrated on the topologically trivial systems in the conventional meaning, the exceptional edge modes also exist in the topologically nontrivial systems. To demonstrate the appearance of the exceptional edge modes in nontrivial systems and their robustness against disorders, we have done additional calculations and added discussions in the manuscript. We have also realized that the model considered in the previous

manuscript can actually be nontrivial in the sense that the pseudo-Hermiticity can protect the edge modes. We have revised the models to show the existence of exceptional edge modes in genuine trivial systems (in the conventional sense). Below we would like to explain this point in more detail.

Additional Figure 4. Edge band structures of the nontrivial model under disorders. (a) Exceptional edge modes appear in the nontrivial model without disorders. (b) Exceptional edge modes still exist under the random real on-site potential and the imaginary noise in the non-Hermitian coupling. (c) Gapless edge modes disappear under the real noise in the non-Hermitian coupling. (d) Gapless edge modes are recovered by the imaginary on-site potential even under real noise in the non-Hermitian coupling.

To show the existence of exceptional edge modes in nontrivial systems, we consider the following non-Hermitian Bernevig-Hughes-Zhang model:

$$H(\mathbf{k}) = (u + \cos k_x + \cos k_y)I_2 \otimes \sigma_z + \sin k_y I_2 \otimes \sigma_y + \sin k_x \sigma_z \otimes \sigma_x + i\beta \sigma_x \otimes \sigma_x.$$

One can confirm nontrivial \mathbb{Z}_2 and \mathbb{Z} invariants protected by the time-reversal symmetry and the pseudo-Hermiticity of this model. To confirm the robustness of the edge modes, we numerically calculate the edge band structures under disorders similar to those considered in Fig. 3: a random real on-site potential, imaginary and complex noises in the non-Hermitian coupling, and an additional imaginary on-site potential. The figure above exhibits the edge band structures with and without these disorders and perturbations. We obtain a similar result to Fig. 3 in the main text and confirm that the discussion using the effective edge Hamiltonian is also valid for exceptional edge modes in nontrivial systems. We note that a random real on-site potential and an imaginary noise in the non-Hermitian coupling break the time-reversal symmetry and the pseudo-Hermiticity, which implies that the robustness of the exceptional edge modes is independent of the bulk topological invariant. Please see the revised Supplementary Information for the details of the symmetries and the disorder terms.

We have noticed that the two-layered non-Hermitian Bernevig-Hughes-Zhang model and the model for the topological insulator laser also have nontrivial \mathbb{Z} invariants regarding the pseudo-Hermiticity. To show the existence of the exceptional edge modes in truly trivial systems, we have modified the models and confirmed that we obtain the same conclusions as those in the previous

manuscript. More specifically, we consider the two-layered non-Hermitian Bernevig-Hughes-Zhang model with the Hermitian coupling considered in Fig. 2 and the topological laser model with an imbalanced non-Hermitian coupling. Please see the revised manuscript and Figures for details. We have revised Figures and sentences and added Supplementary Figure to clarify these points as summarized in (26), (27), (S6), and (S7).

“23. In the methods, the authors mention they have use the fourth-order Runge-Kutta method. Could the authors give some details concerning the method in the supplementary information?”

In the Runge-Kutta simulation, we set the time step $dt = 0.001$. We use the parameter in QWZ model $u = -1$ in all cases and the non-Hermitian coupling $\beta = \beta' = 0$ in Fig. 5a, $\beta = 0.2$, $\beta' = 0.1$ in Fig. 5b, and $\beta = 0.9$, $\beta' = 0.8$ in Fig. 5c. We have added sentences to clarify these points as summarized in (28).

“24. In the caption of Fig.1, the authors write that “[f]our gapless bands per edge exist in the bulk energy gap”. This means that we should see eight gapless bands in the real part of the spectrum in 1(b), but instead we see four. Are the bands doubly degenerate?”

As mentioned by the reviewer, the edge bands in Fig. 1 doubly degenerate. We have revised the manuscript to clarify this point as summarized in (29).

“25. At the end of the caption of Fig.1, the authors write “[t]hese EPs play the role of the ‘glue’ of the edge modes and thus prevent gap opening[s] by perturbations or disorders”. To highlight this latter point, they could add to this sentence that this is shown in Fig.3.”

We have implemented this change suggested by the reviewer in the revised manuscript as described in Summary of changes made (30).

“26. In Fig.3, the insets of the figures show exceptional points. Firstly, which exceptional points are shown in this inset with regards to the main figures? Secondly, do these exceptional points still exist in Figs.3(c) and (d) because from the main figures it seems as if they have disappeared? Thirdly, could the authors perhaps better explain what they are showing in the insets and what it means (see also the next comment)?”

First, all the exceptional points in the main figures correspond to the exceptional points in the insets. The most important ones are those around $E = 0$ because those gapless points are not protected by the conventional topology. Second, in the main figure, we concentrate on the real wavenumber $\text{Im}k_y = 0$ due to the periodic boundary condition in the y-direction, and thus exceptional points disappear in the main figure of Figs. 3c and 3d. However, if we consider the complex wavenumber, there still exist exceptional points in the edge band structures as shown in the inset. Finally, we find that we can better explain the meaning of insets and thus have revised the caption (please see also the reply to the next comment). We have also added discussions to clarify these points as summarized in (31).

“27. At the end of the caption of Fig.3, it says “the edge modes are recovered even under the random Hermitian couplings because the real axis of the wavenumber plane crosses the degeneracy curve of the real parts of the eigenenergies”. Is this the case because k_y should be real?”

As the reviewer points out, the recovery of edge modes by the imaginary on-site potential is due to the constraint that k_y should be real under the periodic boundary condition in the y-direction. We have revised discussions to clarify this point as summarized in (32).

“28. In the real part of the spectrum shown in Fig.6(b), several band crossings can be observed for the in-gap states, of which two correspond to exceptional points and four to normal band crossings. However, there are two more crossings on the $\text{Re } E = 0$ axis. What do these correspond to? Also, if the four band crossings indicated by the green triangles are not really band crossings because they avoid each other in the imaginary parts of the energy, then what is the circular band exactly? Can it for example be pushed out of the gap into the bulk bands?”

Band crossings on the $\text{Re } E = 0$ axis which are not depicted by the red circles are the points where unstable edge bands appear from the bulk bands around $\text{Re } E = 0$. Thus, those crossings do not disappear as long as those unstable edge bands separately appear from bulk bands. For example, pushing up the circular band into the upper bulk bands is prohibited as demonstrated in the figure below. On the contrary, we can modify the band structures so that the circular band overlaps the bulk bands on the $\text{Re } E = 0$ axis. We can also remove the apparent crossing depicted by the green triangles in Fig. 6 (Fig. 7 in the revised manuscript) by the modification like in the figure below. We have added discussions to clarify this point as summarized in (33) and (S8).

Additional Figure 5. Possible and impossible deformation of the edge band structure in the chiral active matter model.

“29. As a general comment to all the figures in the main text (Figs.1-6), in none of the caption of the figures it is mentioned explicitly for which parameters the plots are made. To relate the figures to the different terms discussed in the main text, it would be helpful to include this information.”

We have added sentences to specify the parameters used in the calculations for the caption of each figure. This is described in Summary of changes made (34).

“30. In the supplementary information it is shown on page 2 why the bulk energy of the Hamiltonian in Eq. (1) in the main text has zero imaginary part by making use of the Bloch Hamiltonian. However, as bulk-boundary correspondence is broken, how can the authors be sure that results computed from the Bloch Hamiltonian have any merit when taking open boundary conditions?”

We thank the reviewer for this insightful comment. We have done additional calculations to confirm that the bulk eigenstates are similar to those predicted from the Bloch Hamiltonian. The figure below shows an example of the bulk eigenstates under the open-boundary condition in the x -direction and the periodic boundary condition in the y -direction. The shape of the bulk eigenstates is similar to the Bloch wave (i.e., sine curve). Furthermore, the bulk eigenstates are localized at either former (1-4) or latter (5-8) part of the sublattice, which implies that the eigenstates of the two-layered QWZ model and its time-reversal are separated even under the small couplings as predicted in our discussion in the Supplementary Information. These results should justify the use of the Bloch Hamiltonian to discuss the bulk eigenstates.

Additional Figure 6. Example of bulk eigenmodes in the two-layered non-Hermitian Bernevig-Hughes-Zhang model.

The breakdown of the bulk-edge correspondence mentioned in the manuscript indicates the existence of the exceptional edge modes that cannot be predicted by the bulk topology. Except for the existence of the exceptional edge modes, our model can exhibit the conventional bulk-edge correspondence. In the present consideration, the known (other) mechanism of the breakdown of the bulk-edge correspondence, such as the non-Hermitian skin effect, is irrelevant. Thus, the bulk eigenstates can be predicted by the Bloch Hamiltonian as in the Hermitian systems (as also demonstrated above).

Furthermore, we can justify the use of Bloch Hamiltonian and its eigenvectors from the Hermiticity of the nonperturbed Hamiltonian without the non-Hermitian coupling. We also discuss the PT symmetry of the Hamiltonian, which ensures that the eigenvalues are real or appear as pairs of complex conjugates. Please see Supplementary Information in the revised manuscript for details. We have added discussions to clarify these points as summarized in (S9).

“31. Could the authors specify why Figs.4 and S1 are different as they seem to be plotted for the same model? Is it simply different parameter settings or are they plots for different models?”

Figures 4 and S1 (S3 in the revised manuscript) represent the edge band structure for different models. Figure 4 shows the band structure of the model with two layers of the QWZ model in Eq. (5). On the other hand, Figure S1 (S3 in the revised manuscript) is about the model with four layers of the QWZ model represented in Eq. (S19) in the revised manuscript and thus has more edge bands than that in Fig. 4. We have added discussions to clarify these points as summarized in (S10).

“32. To which model in the main text does the continuum model in Eq.(S16) in the supplementary information correspond?”

The model mentioned by the reviewer corresponds to the argument in the first paragraph in the section “Active matter realization of exceptional edge modes”, which refers to the existence of exceptional edge modes in continuum systems. We have implemented the figure of the band structure of the continuum model in the main text.

In the process of revising the manuscript, we have realized that the continuum model analyzed in the previous manuscript can actually have a nontrivial topological invariant under the pseudo-Hermiticity condition. Thus, we have modified the continuum model into a genuine trivial one. Particularly, we use a different type of non-Hermitian coupling. We numerically confirm that the revised continuum model exhibits exceptional edge modes even without the pseudo-Hermiticity. We also check that the sum of the Chern numbers of the bulk bands below the energy gap is zero, and thus this continuum model has a topologically trivial bulk in the conventional meaning. Please see the revised manuscript for the details of the revised model and the numerical result. We have

added a figure in the main text and revised discussions to clarify this point and to improve the readability as summarized in (35) and (S11).

Once again, we are very grateful to Reviewer #2 for a positive appreciation of our work and for giving many interesting and useful suggestions, which we find have been very helpful in improving the manuscript. We have revised the manuscript to meet all the requirements for further clarifications on the arguments presented in the manuscript. We do hope that the revised manuscript, together with our reply above, will meet with the reviewer's approval.

REVIEWERS' COMMENTS

Reviewer #1 (Remarks to the Author):

Thank the authors for their detailed responses. I think the discussion based on symmetries is very meaningful. My concerns about robustness and novelty are well addressed. On the application side, enough details are presented now. Based on these points, I think this manuscript can meet the standard of Nature Communications

Reviewer #2 (Remarks to the Author):

The authors have greatly improved the clarity and legibility their work, and more clearly highlighted their interesting findings.

I have some additional questions and comments for the authors:

1. In their response to my first comment, the authors make a distinction between the bulk-boundary correspondence and the bulk-edge correspondence by equating the first as applying to boundaries separating regions with different topological invariants and the latter as applying to boundaries surrounded by the vacuum. To my knowledge, the bulk-boundary correspondence is used to refer to both situations. However, as this is a matter of semantics, I do not insist on this point.
2. In response to my third comment, the authors write that the bulk-boundary correspondence has only been reestablished in the non-Hermitian context in one-dimensional systems with chiral symmetry and two-dimensional systems with no symmetries. While this is true based on the examples studied in the literature, I would like note that several of the methods introduced to redefine the bulk-boundary correspondence, for example, the work in Ref.[25], can be straightforwardly generalised to models of any dimension with any symmetry.
3. In lines 48-50 in the main text ("In one-dimensional ... chiral symmetry.") and lines 162-163 (In one-dimensional ... chiral symmetry."), the authors refer to the emergence of exceptional points in the presence of certain symmetries and cite Ref.[40]. I would like to point out that Ref. [58] and Okugawa and Yokoyama (Phys. Rev. B 99, 041202 (R) (2010)) were the first works to study this.
4. There is a typo in line 143: It should read " $\text{Re } \beta \neq 0$ " instead of " $\text{Re } \beta = 0$ ".
5. I am wondering what the added benefit of introducing Eq.(4) in the main text is. Almost immediately afterwards another Hamiltonian is introduced for which plots are shown. Is there a reason why Eq.(4) is important?
6. At the bottom of page 22 and top of page 23, the authors refer to Fig. S6, I believe this should be Fig. S7.

Reply to Reviewer #1

We are very grateful to Reviewer #1 for her/his second-round review and the recommendation of publication in Nature Communications.

Reply to Reviewer #2

We are very grateful to Reviewer #2 for her/his careful reading of our manuscript and a positive appreciation of our work. Reviewer #2 also made some specific comments, which we find have been helpful to improve the correctness and readability of our manuscript. Here, let us reply to each of them.

“The authors have greatly improved the clarity and legibility their work, and more clearly highlighted their interesting findings.

I have some additional questions and comments for the authors:”

We thank the reviewer for a positive appreciation of our results and our efforts in revising the manuscript. We address each comment raised by the reviewer below.

“1. In their response to my first comment, the authors make a distinction between the bulk-boundary correspondence and the bulk-edge correspondence by equating the first as applying to boundaries separating regions with different topological invariants and the latter as applying to boundaries surrounded by the vacuum. To my knowledge, the bulk-boundary correspondence is used to refer to both situations. However, as this is a matter of semantics, I do not insist on this point.”

We agree with the reviewer that the bulk-boundary correspondence can indicate the situation that edge modes appear at the boundary between a topological system and the vacuum. In our opinion, however, the breakdown of the bulk-boundary correspondence should also imply the appearance of robust edge modes at the boundary of two systems with the same bulk topological invariants as well as those in the system surrounded by vacuum. Since we have not analyzed such a situation, we still choose to use the terminology of bulk-edge correspondence to describe the scope of our study.

“2. In response to my third comment, the authors write that the bulk-boundary correspondence has only been reestablished in the non-Hermitian context in one-dimensional systems with chiral symmetry and two-dimensional systems with no symmetries. While this is true based on the examples studied in the literature, I would like note that several of the methods introduced to redefine the bulk-boundary correspondence, for example, the work in Ref.[25], can be straightforwardly generalised to models of any dimension with any symmetry.”

We tend to agree with the referee that the previous methods to redefine the bulk-boundary correspondence can be extended to non-Hermitian systems with arbitrary dimensions and symmetries. On the other hand, in our understanding, there are only few explicit arguments that guarantee the validity of such generalization. This is the reason why we considered that the bulk-edge correspondence in non-Hermitian systems is less established compared to that in the Hermitian case. Overall, however, we have realized that the description of the bulk-edge correspondence in the previous manuscript may be excessive, and thus have revised sentences as follows [Summary of changes (1)].

“However, the bulk-edge correspondence is more subtle in non-Hermitian systems than in Hermitian systems, as unexplored non-Hermitian effects may protect unpredicted edge modes or remove edge modes from topologically nontrivial systems.”

“3. In lines 48-50 in the main text (“In one-dimensional ... chiral symmetry.”) and lines 162-163 (In one-dimensional ... chiral symmetry.”), the authors refer to the emergence of exceptional points in the presence

of certain symmetries and cite Ref.[40]. I would like to point out that Ref. [58] and Okugawa and Yokoyama (Phys. Rev. B 99, 041202 (R) (2010)) were the first works to study this.”

We thank the reviewer for this helpful comment. We have added the references raised by the reviewer to acknowledge the early work of symmetry-protected exceptional points [Summary of changes (2)].

“4. There is a typo in line 143: It should read “ $\text{Re } \beta \neq 0$ ” instead of “ $\text{Re } \beta = 0$.”

We thank the reviewer for this useful comment. We have corrected the sentence as described in Summary of changes (3).

“5. I am wondering what the added benefit of introducing Eq.(4) in the main text is. Almost immediately afterwards another Hamiltonian is introduced for which plots are shown. Is there a reason why Eq.(4) is important?”

We have introduced the effective theory in Eq. (4) to provide a guiding principle to construct lasing edge modes with the nonzero group velocity independently of details about the microscopic Hamiltonian. Equation (4) should also lead to a better understanding of the mechanism of exceptional edge modes with nonzero group velocity in the model described by Eq. (5). Specifically, we can derive the dispersion relation $E = E_0 + (a - 1)k \pm \sqrt{(a + 1)^2 k^2 - 4\beta\beta'}$ from Eq. (4), which indicates the existence of exceptional points and the nonzero group velocity. We have revised and added sentences to clarify this point as follows [Summary of changes (4)].

“Specifically, we find that the general form of effective edge Hamiltonians is given by...”

“We derive the dispersion relation of this effective Hamiltonian, $E = E_0 + (a - 1)k_y \pm \sqrt{(a + 1)^2 k_y^2 - 4\beta\beta'}$, which exhibits exceptional points at $k_y = \pm 2\sqrt{\beta\beta'}/(a + 1)$ and nonzero group velocity.”

“6. At the bottom of page 22 and top of page 23, the authors refer to Fig. S6, I believe this should be Fig. S7.”

As a matter of fact, Fig. S6 is correct in the sentences raised by the reviewer. There, we mention Supplementary Fig. 6 to show an example of band structures that exhibit no exceptional edge modes but still preserve conventional PT and CP symmetry and modified pseudo-Hermiticity. However, we found that the manuscript was indeed confusing. We have revised those sentences as follows [Summary of changes (S1)].

“We can also check that the imaginary on-site potential considered to calculate the band structures in Fig. 3d in the main text and in Supplementary Fig. 6 preserves...”

“The disappearance of exceptional edge modes from the band structures in Fig. 3d in the main text and in Supplementary Fig. 6 indicates...”

Once again, we are very grateful to Reviewer #2 for a positive appreciation of our work and for giving many useful comments. We have revised the manuscript to meet all the requirements for improving the clarity and the fairness of the manuscript. We do hope that the revised manuscript, together with our reply above, will meet with the reviewer’s approval.